# Detection of DNA base modifications by deep recurrent neural network on Oxford Nanopore sequencing data

Qian Liu[1], Li Fang [1], Guoliang Yu[2,3], Depeng Wang[3], Chuan-Le Xiao [2] & Kai Wang[1,4]

DNA base modifications, such as C5-methylcytosine (5mC) and N6-methyldeoxyadenosine (6mA), are important types of epigenetic regulations. Short-read bisulfite sequencing and long-read PacBio sequencing have inherent limitations to detect DNA modifications. Here, using raw electric signals of Oxford Nanopore long-read sequencing data, we design Deep-Mod, a bidirectional recurrent neural network (RNN) with long short-term memory (LSTM) to detect DNA modifications. We sequence a human genome HX1 and a *Chlamydomonas reinhardtii* genome using Nanopore sequencing, and then evaluate DeepMod on three types of genomes (*Escherichia coli*, *Chlamydomonas reinhardtii* and human genomes). For 5mC detection, DeepMod achieves average precision up to 0.99 for both synthetically introduced and naturally occurring modifications. For 6mA detection, DeepMod achieves ~0.9 average precision on *Escherichia coli* data, and have improved performance than existing methods on *Chlamydomonas reinhardtii* data. In conclusion, DeepMod performs well for genome-scale detection of DNA modifications and will facilitate epigenetic analysis on diverse species.

---

[1] Raymond G. Perelman Center for Cellular and Molecular Therapeutics, Children's Hospital of Philadelphia, Philadelphia, PA 19104, USA. [2] State Key Laboratory of Ophthalmology, Zhongshan Ophthalmic Center, Sun Yat-sen University, Guangzhou 510060, China. [3] Grandomics Biosciences, Beijing 102200, China. [4] Department of Pathology and Laboratory Medicine, Perelman School of Medicine, University of Pennsylvania, Philadelphia, PA 19104, USA. Correspondence and requests for materials should be addressed to C.-L.X. (email: xiaochuanle@126.com) or to K.W. (email: wangk@email.chop.edu)

DNA base modifications are modified versions of standard nucleotides in a DNA molecule through the addition of chemical groups. For example, DNA methylations, including 5-methylcytosine (5mC), 5-hydroxymethylcytosine (5hmC), and N6-methyldeoxyadenosine (6mA), are introduced into a DNA molecule by adding methyl or hydroxymethyl groups to nucleotides: 5mC is introduced by adding methyl group at the 5th position of the pyrimidine ring of cytosines, whereas 6mA is introduced by adding methyl group at the 6th position of the purine ring in adenines. There are also other different types of methylations, named according to the nucleotide type, the added molecule type and the modified position in nucleotides, such as 4mC (N4-methylcytosine), 5hmC (5-hydroxymethylcytosine), 5fC (5-formylcytosine), and 5caC (5-carboxylcytosine) introduced by methyltransferases[1], and they generally do not disrupt base pairing. Some other modifications, such as 1mA (N1-methyladenine), 3mA (N3-methyladenine), 7mA (N7-methyladenine), 3mC (N3-methylcytosine), 2mG (N2-methylguanine), 6mG (O6-methylguanine), 7mG (N7-methylguanine), 3mT (N3-methylthymine), and 4mT (O4-methylthymine), may damage hydrogen bonds in base pairs[1,2]. DNA base modifications widely exist in different organisms[3] and are essential in various biological processes[4,5], such as genomic imprinting, X-chromosome inactivation, genome stability, gene regulation, repression of transposable elements, aging, and carcinogenesis. For example, hyper-methylation of 5mC in promoter regions usually represses gene transcription, and thus may regulate cellular processes such as cell differentiation and pluripotency. Local DNA 5mC hyper-methylation and genome-wide hypo-methylation have been seen in cancer[6], and the differential methylation of CpG islands can distinguish cancer cells from normal cells[7] or different tumor types[8], and thus may be a potential cancer biomarker or a therapeutic target.

DNA modifications, especially DNA methylations, could be detected by both short-read sequencing and long-read sequencing techniques. Bisulfite sequencing with short-read techniques is widely used to call methylated cytosines[9] by converting unmethylated cytosines to uracil, but bisulfite sequencing and its improved variations[10] were limited by the efficiency in conversion and the inability to assay repetitive genomic regions by short-read sequencing. Immunoprecipitation followed by short-read sequencing can detect DNA or RNA modifications[11,12] in genomic regions but it lacks single-base resolution. PacBio long-read sequencing can be used to directly detect DNA/RNA modifications[13] based on altered polymerase kinetics during sequencing[14–19]. However, this method has low signal-to-noise ratio for 5mC modifications[20], requires relatively high coverage for calling modifications, and is biased by incomplete and context-dependent enzymatic treatment of 5mC detection using Tet1[21].

Recently, several proof-of-concept studies on Oxford Nanopore sequencing techniques have demonstrated the feasibility to detect DNA modifications based on the electric signal characteristics when a modified DNA molecule passes through nanopores[22,23]. DNA methylations at specific genomic positions can be measured with higher accuracy by comparing raw electric signals of methylated DNA copies with signals of the same unmethylated DNA copies[22–25] for known sequences without large prior training data set[3]. Furthermore, DNA methylation detection using Nanopore sequencing was used in a few studies[4,26,27] with the help of machine learning algorithms. Simpson et al. developed a HMM (hidden Markov model) to measure long reads with complete methylations and long reads without methylations separately, and used the log-likelihood ratio to distinguish 5mC from cytosine in *Escherichia coli* and *Homo sapiens*[4]. Similarly, Rand et al. used HMM with a hierarchical Dirichlet process to identify three types of cytosine methylations (i.e., cytosine, 5mC and 5hmC) and also 6mA in *E. coli* with different phases[20]. McIntyre et al. further improved the detection of 6mA (implemented in mCaller) in mouse, *E. coli* and Lambda phage DNA[26] using signal deviation of six 6-mer around positions of interest as input.

However, the sequential characteristic of Nanopore electric signals was not fully utilized in previous studies, and the performance of 5mC/6mA prediction may be improved using more sophisticated deep neural networks. In the current study, we developed a computational tool, DeepMod (Fig. 1), which takes a reference genome and long-read electric signals together with

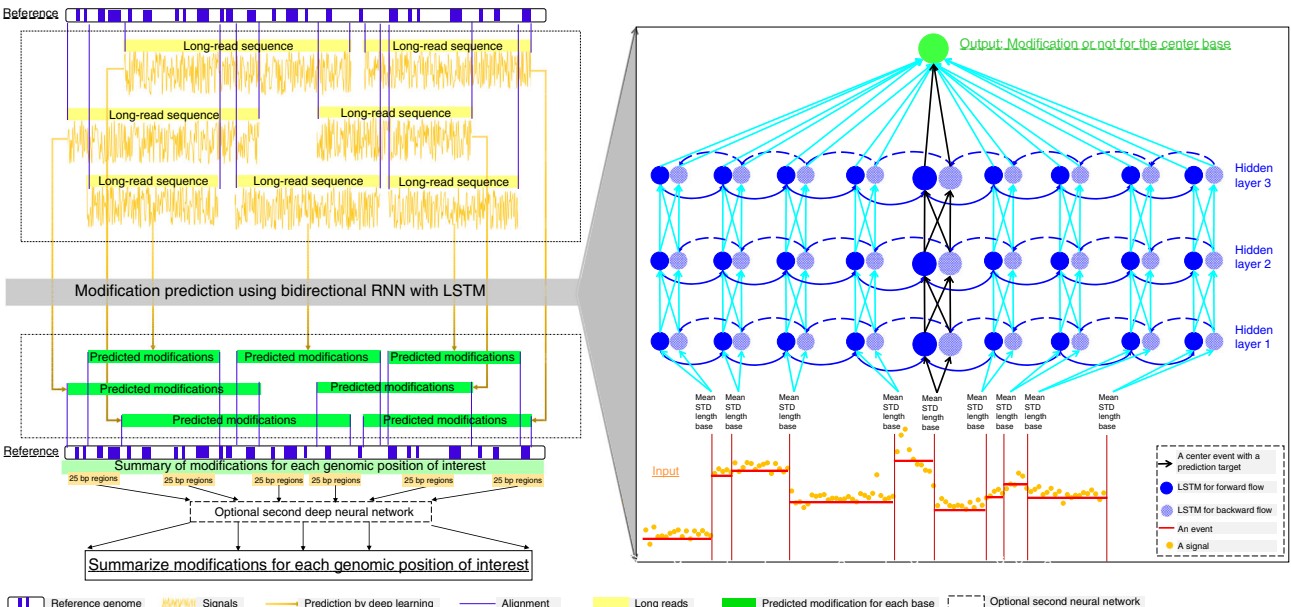

**Fig. 1** The flowchart of DeepMod. RNN: recurrent neutral network; LSTM: long short-term memory. Several long reads (in yellow) were shown for demonstration purposes with their alignment and Nanopore signals (yellow lines). To view LSTM RNN for modification prediction, nine LSTM cells adjacent to the center position (with arrows in black) were shown

event information generated from Nanopore sequencing as input, and outputs modification summary for genomic positions of interest in a reference genome, together with modification prediction for bases of interest in a long read. The modification prediction model in DeepMod is a well-trained bidirectional recurrent neural network (RNN) with long short-term memory (LSTM)[28] units, which takes signal mean, standard deviation, and the number of signals of an event together with base information in the reference genome of an event and its neighbors as input, and makes modification prediction for the event. Then, after anchoring events with a reference genome based on the alignment of long reads, predicted modification summary for reference positions of interest can be generated in a BED format, indicating how many reads cover each genomic position and how many reads contain predicted modifications at genomic positions of interest. The prediction of DNA modification by DeepMod is thus strand-sensitive and has single-base resolution. We note that one recent preprint[29] also used the RNN framework (DeepSignal) but with limited evaluations. In the current study, we sequence a human genome HX1 and a *Chlamydomonas reinhardtii* genome using Nanopore sequencing techniques, and together with published Nanopore data for *Escherichia coli* and another human genome NA12878, we evaluate DeepMod on three types of genomes (*E. coli*, *C. reinhardtii*, and human genomes) and show that it performs well on genome-scale detection of DNA modifications.

## Results

**A brief summary of the DeepMod algorithm.** We designed DeepMod to capture time-series characteristic of Nanopore signals for detecting DNA modifications. DeepMod takes FAST5 files and a reference genome as inputs, where the FAST5 files are generated from Nanopore sequencer and must include event information after basecalling. The output files of DeepMod include BED files with coverage and methylation percentage information for genomic positions of interest. DeepMod works through several steps as illustrated in Fig. 1.

First, a sequence of bases is extracted from events in each FAST5 file, and then aligned with a reference genome. Second, the raw signals of the events and the mapped reference base type of the events is obtained, and the summary of the raw signals of each of $w$ events (by default, $w = 21$) with the reference base type of events (A or C or G or T) is used as input of a LSTM RNN to predict whether the center event of the $w$ events is generated from a modified base (the left part of Fig. 1). After repeating the prediction process by the LSTM RNN for events of interest in a long read and then for all long reads, the sequence coverage and the methylation coverage are generated for genomic positions of interest in the reference genome. The whole-genome modification summary is formatted as BED files for performance evaluation and other analysis. In addition, a second neural network is also used for some types of modifications, such as 5mC[30] to incorporate high correlation of methylation of a CpG site and its nearby CpG sites. The second neural network takes the predicted methylation percentage of a reference position of interest in the genome and the methylation percentage of its neighboring sites in both strands as input and generates a new methylation percentage of the position of interest. In this study, DeepMod was evaluated on three types of genomes with two types of modifications (5mC and 6mA), and the performance evaluations were described in detail below.

**Performance of 5mC detection on Nanopore sequencing data on *E. coli*.** DeepMod was first trained on a published Nanopore sequencing data on *E. coli* with positive and negative controls[4] (See the Methods section and Supplementary Notes). The negative control (UMR for short) were amplified by polymerase chain reaction (PCR) and thus the reads contain no modifications; in comparison, reads in the positive control (CG_MSssI for short) were treated by the M.SssI methyltransferase after PCR amplification and thus the majority of CpG sites were methylated[2]. The summary of the two data sets was described in Table 1. To select suitable hyper-parameters in DeepMod for the tradeoff between prediction performance and running time/consuming resources, two independent validation strategies were used on UMR and CG_MSssI. The first strategy is read-based-independent validation, where 90% of the reads from the negative control and 90% of the reads from the positive control were randomly selected and used to train DeepMod, whereas the rest (10%) of the reads were used for testing. Under this validation strategy, reads in the training and testing groups might be aligned to the same genomic regions. In parallel, in the second region-based independent validation strategy, reads mapped to the genomic positions from 1,000,000 to 2,000,000 of *E. coli* reference genome were used for testing, whereas reads mapped to other genomic regions were used for training. We also evaluated two additional hyper-parameters in DeepMod: different $w$ (i.e., the number of LSTM units ranging from 7 to 51), 7-feature description of an event

---

**Table 1 Nanopore sequencing data sets used to evaluate DeepMod**

| Genome | Data set name | Motif | # reads | Coverage | Meth[a] | Reference |
|---|---|---|---|---|---|---|
| *Escherichia coli* | UMR | NA[d] | 111,238 | 110X | Neg[b] | Simpson et al. [4] |
| | CG_MsssI | CG[c] | 69,899 | 67X | 5mC | |
| | CG_SssI | CG[c] | 8679 | 19X | 5mC | |
| | CG_MpeI | CG[c] | 23,593 | 39X | 5mC | |
| | GCGC_HhaI | GCGC[c] | 18,180 | 50X | 5mC | |
| | gaAttc_EcoRI | GAATTC[c] | 16,661 | 27X | 6mA | Stoiber et al. [3] |
| | gAtc_dam | GATC[c] | 17,557 | 33X | 6mA | |
| | tcgA_TaqI | TCGA[c] | 16,249 | 22X | 6mA | |
| | Con1 | NA[d] | 23,762 | 34X | Neg[b] | |
| | Con2 | NA[d] | 34,170 | 40X | Neg[b] | |
| *Homo sapiens* | NA12878 | CG[c] | | 30X | 5mC | Jain et al. [31] |
| | HX1 | CG[c] | 4,827,155 | 30X | 5mC | Current study |
| *Chlamydomonas reinhardtii* | C. reinhardtii | NA[d] | 772,817 | 126X | 6mA | Current study |

[a] Methylation types
[b] Negative control without any modifications
[c] Underlined nucleotides in motifs were potential modified target.
[d] No modifications or no motif information

(base information, signal mean, standard deviation, and the number of signals associated with an event) and 57-feature description of an event (7-features description plus 50 discretized bins of signals). The performance was measured using accuracy, precision, recall, and F1-score (See Eqs. (3–6) in the Methods section for their definitions). The results of the two independent validation above were shown in Supplementary Table 1 for per-call performance (where the prediction of a base in each long read was considered separately without alignment). We found that (i) 57-feature description produced similar performance to 7-feature description, and (ii) larger $w$ generated increasing accuracy, recall and F1-score, but the memory usage and running time were much higher beyond $w = 21$. We thus used 7-feature description and $w = 21$ as the default setting in DeepMod. Cross-validation of region-based independent validation strategy was also provided in Supplementary Table 2 for DeepMod with 7-feature description and $w = 21$, and the performance values in five different regions ([0, 1000000], [1000000, 2000000], [2000000, 3000000], [3000000, 4000000], [4000000, 4700000]) were similar to each other.

DeepMod's cross-data set performance was further evaluated on another independent *E. coli* Nanopore data set[3] where there were two negative control samples without any modifications (denoted as Con1 and Con2 for short), and three positive control samples with 5mC modifications. On the three positive control samples, three methylases (M.Hhal, M.Mpel, and M.Sssl) were used separately to synthetically introduce 5mC for 70,180 GCGC motifs (the underlined bases represents modified cytosines by enzymes hereafter) and 693,586 CG motifs in *E. coli*, and thus the samples were denoted as GCGC_Hhal, CG_MpeI, and CG_SssI for short. DeepMod was used to make the modification predictions on the five Nanopore sequencing data sets (the three positive control data with 5mC and two negative control without modifications). We mixed each of three positive control data with both Con1 and Con2 for performance evaluation of DeepMod.

There were two types of evaluations on this independent *E. coli* data set. One type of evaluation was based on methylated cytosines within specific sequence motifs of interest. Under this type of evaluation, the number of the cytosines of interest in motifs from positive control is roughly equal to that from Con1 and Con2 as shown in Table 2. In Table 2, there are ~ 1.4 million CpG sites and ~ 0.14 million GCGC sites, where about half were 5mC from positive control and the other half have no modification from the negative control. The performance of DeepMod was then evaluated using AUC and AP (AUC: area under curve, and AP: average precision. See more details in Methods). The performance of DeepMod was shown in Fig. 2a and the confusion matrices were provided in Supplementary Table 3. It can be seen from Fig. 2a that DeepMod achieved AP values of 0.990, 0.921 and 0.993, and AUC values of 0.987, 0.906, and 0.988 for GCGC_Hhal, CG_MpeI, and CG_SssI, respectively, for all motif sites of interest (Fig. 2a) for coverage ≥ 1. In particular, given a threshold of prediction methylation percentage ≥ 0.1 for a genomic position of interest, DeepMod gave precision

= 0.945 and recall = 0.97 for GCGC_Hhal, precision = 0.835 and recall = 0.879 for CG_MpeI and precision = 0.848 and recall = 0.986 for CG_SssI. As CG_SssI (19X) has much lower coverage than GCGC_Hhal (50X) and C G_MpeI (39X), if the threshold of prediction methylation percentage was set to a larger value for CG_SssI, i.e., ≥ 0.2, DeepMod achieved precision = 0.96 and recall = 0.985 with significant improvement of precision but a slight (0.001 point) decrease of recall. Meanwhile, a larger coverage threshold such as 5 can yield better performance, although the improvement is not significant as shown in Fig. 2a.

Besides the evaluation on methylated cytosines within sequence motifs of interest above, the second type of evaluation on the independent *E. coli* data were based on all cytosines from both positive control and negative control, and the results were illustrated in Fig. 2b, c: DeepMod achieved AUC of 0.985, 0.953, and 0.992 for all cytosines in the GCGC_Hhal data set, the CG_MpeI data set, and the CG_SssI data set, respectively, suggesting similar AUC values when moving the classification of motif (GCGC or CG)-based cytosines to the classification of all cytosines. However, from Fig. 2c, the AP values of DeepMod to classify all cytosines are 0.771, 0.890, and 0.983 in the GCGC_Hhal data set, the CG_MpeI data set and the CG_SssI data set, respectively, which were lower than the performance for the prediction of cytosines in sequence motifs, indicating misclassifications for cytosines that are not present in motifs. When we checked which cytosines might be more likely to be misclassified, we found that cytosines closely adjacent to modified cytosines (i.e., one or two upstream/downstream cytosines) has much higher likelihood to be misclassified than cytosines in other positions, suggesting possible neighborhood effect caused by modifications. In fact, the misclassification of cytosines adjacent to modified cytosines may not be wrong but merely indicates the existence of modified cytosines that are adjacent to each other, and thus the prediction power of our method may be under-estimated in these scenarios. In summary, this analysis on several independent data sets clearly demonstrated the satisfactory performance of DeepMod in genome-wide, single-base resolution detection of 5mC.

**Performance of 5mC prediction on independent Nanopore sequencing data on human genomes.** Cross-species 5mC prediction performance of DeepMod was further evaluated on a human Nanopore sequencing data set. Under this evaluation, a DeepMod model, which was trained on *E. coli* data, were used to make 5mC prediction on the NA12878 human cell line. Recently, NA12878 was sequenced by Jain et al. mainly using Nanopore R9.4 with ~ 30X coverage[31]. On this human Nanopore sequencing data, the well-trained DeepMod model on *E. coli* data were used to make 5mC prediction for each Nanopore long read, and each cytosine in a long read was associated with a label indicating whether it was predicted to be 5mC or not. Then, after all long reads were aligned against a reference genome, predicted methylation summary for the whole-genome was generated for each cytosine in the genome with a percentage to indicate how

| Table 2 The number of modified and un-modified bases of interest used for evaluation on *E. coli* when coverage ≥ 1 | | | | | |
|---|---|---|---|---|---|
| Data set name | Base of interest | Modification | # modified base in motif | # non-modified base in motif | # base not in motif |
| CG_MpeI | cytosine (C) | 5mC | 693,518 | 693,427 | 3,326,971 |
| CG_SssI | | | 682,526 | 693,427 | 3,297,528 |
| GCGC_Hhal | | | 70,172 | 70,160 | 4,573,777 |
| gaAttc_EcoRI | adenine (A) | 6mA | 277 | 280 | 1,003,885 |
| gAtc_dam | | | 7816 | 7831 | 989,363 |
| tcgA_TaqI | | | 6351 | 6403 | 989,375 |

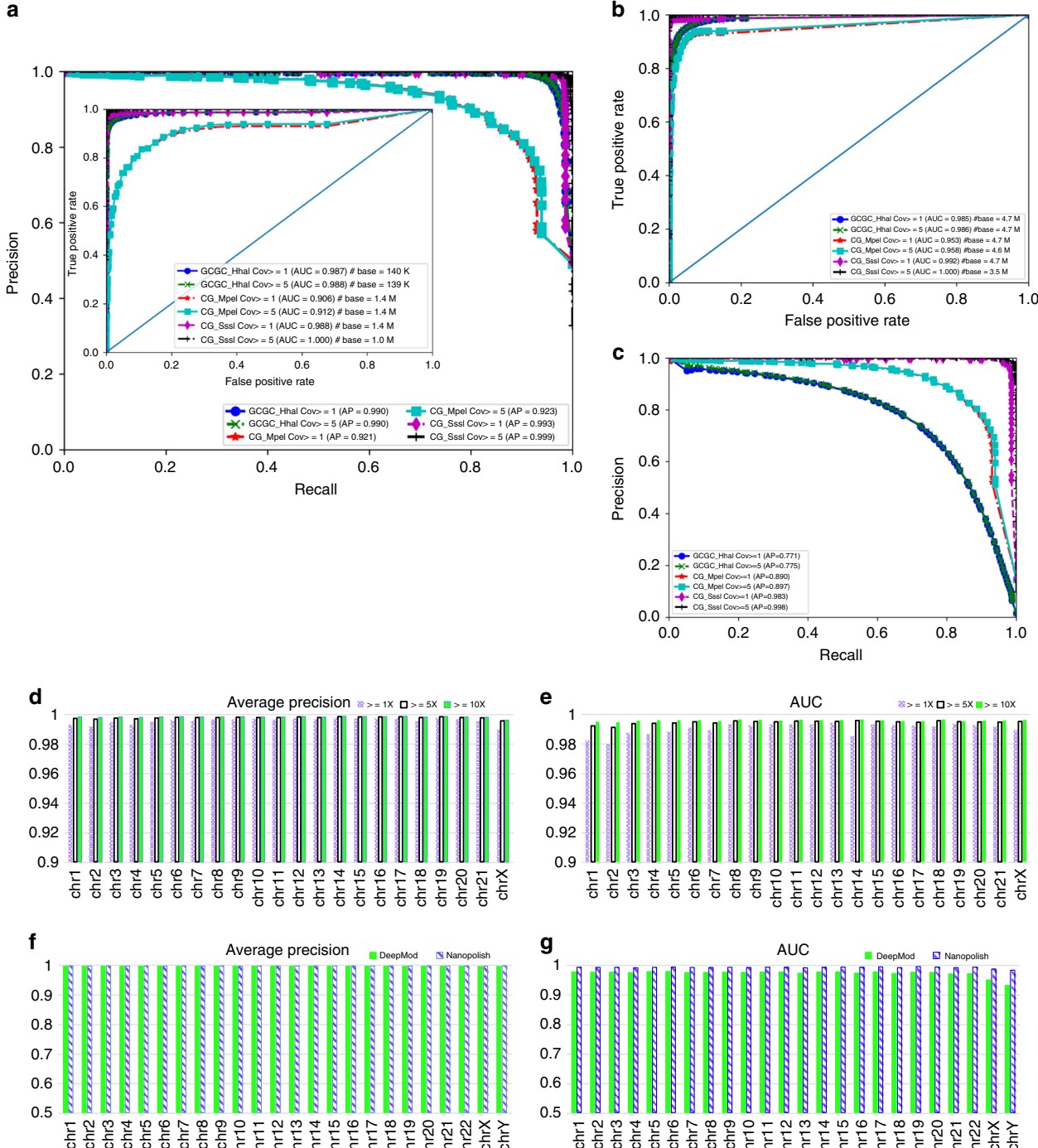

**Fig. 2** Evaluation of the performance of DeepMod on 5mC prediction on *E. coli*, NA12878, and HX1. **a** AP (the outer) and AUC (the inner) plots for 5mC within sequence motifs in *E. coli* for three synthetically introduced 5mC data sets by M.Mpel (CG_Mpel for CG motif), M.SssI (CG_SssI for CG motif), and M.Hhal (GCGC_Hhal for GCGC motif), respectively. **b**, **c** AUC and AP plots for 5mC prediction of all cytosines in *E. coli*. **d**, **e** AP and AUC of 5mC prediction by DeepMod on NA12878. **f**, **g** AP and AUC of 5mC prediction by DeepMod on HX1. Cov: coverage. # base: total number of bases used in the evaluation

many modification predictions were made for that position, compared with the number of reads mapped to that position. Furthermore, to account for the high correlations of nearby 5mC in CpG sites[30], we designed another deep neural network, and the second neural network incorporates the predicted methylation percentages of a CpG site, the prediction on the opposite strand and the methylation prediction of its neighboring CpG sites, and outputs a final prediction percentage for a genomic position. This

second deep neural network could improve AP by 1–3% points and AUC by 3–5% for all chromosomes when coverage ≥ 1.

To evaluate DeepMod prediction performance on NA12878, we used 5mC calls from bisulfite sequencing of NA12878 as benchmark[32]. Owing to the heterogeneity of sequenced samples, we used completely methylated and completely un-methylated bases for the evaluation: a genomic position of a base was considered to be completely methylated if its methylation

percentage ≥ 90% in both replicates of bisulfite sequencing with coverage ≥ $c$ ($c$ could be 1, 5, or 10), and to be completely un-methylated if its methylation is 0% in both replicates. The number of completely methylated and un-methylated bases for each chromosome in NA12878 were given in Supplementary Table 4. We compared methylation percentage of DeepMod prediction against the completely methylated and un-methylated cytosines at reference positions of interest for each chromosome. The results on AP were shown in Fig. 2d, and the AUC was given in Fig. 2e, where both AP and AUC with coverage ≥ 1 were ~0.99, yet when coverage ≥ 5, the values of AP and AUC were > 0.99. For example, on the chromosome 1, when coverage ≥ 5, DeepMod achieved the AP value of 0.998 and the AUC value of 0.993, while on the chromosome 20, when coverage ≥ 5, DeepMod had the AP value of 0.998 and the AUC value of 0.995, indicating accurate prediction on human genome, even when using a DeepMod model trained on E. coli data.

With NA12878 Nanopore data, DeepMod was also evaluated on those 5mC sites where the methylation percentage has small difference (< 0.25) between the two replicates of bisulfite sequencing to account for the heterogeneity of two bisulfite sequencing batches. We found that the prediction methylation percentage has high correlation with the methylation percentage in both replicates (> 0.85 for the majority of chromosomes when coverage ≥ 5). We also noted that DeepSignal[29] trained on E. coli tried to make cross-species prediction of CpG methylation on NA12878, which achieved F1 = 0.949 (with precision = 0.97 and recall = 0.93) on synthetically introduced 5mC in human, whereas nanopolish achieved F1 = 0.89 (with precision = 0.944 and recall = 0.844). Both methods required a CpG site with signals in both the template and complement strands and only output final prediction on template strand by summarizing predictions from both strands, and the prediction was based on synthetically introduced 5mC (complete methylation and un-methylation). Our method on native 5mC prediction in NA12878 achieved F1 = 0.983 (with precision = 0.988 and recall = 0.979) and F1 values on each individual chromosome (calculated based on precision and recall in Supplementary Table 4) were ~ 0.98. Although this is not a direct comparison, DeepMod still demonstrated its highly accurate prediction on NA12878.

DeepMod was further evaluated on another independent human data set HX1, by sequencing whole-blood sample[33] with ~ 30X coverage on the Nanopore platform and by two replicates of bisulfite sequencing of the same sample. To make 5mC prediction on HX1, we trained a model using completely methylated and completely un-methylated cytosines of chromosome 1–10 of NA12878, so that NA12878 and HX1 were basecalled with the same version of Albacore. Then, similar to the evaluation on NA12878, we compared 5mC prediction by DeepMod on Nanopore sequencing data against completely methylated and un-methylated cytosines determined from two replicates of bisulfite sequencing data for HX1 (see Supplementary Table 5 for details). After that, we calculated AP (Fig. 2f) and AUC (Fig. 2g) and precision and recall (Supplementary Table 5) for coverage ≥ 3, where the AP of DeepMod is > 0.99 and the AUC is generally higher than 0.97 except on chromosomes X and Y. The performance was comparable to nanopolish, but both methods have already achieved excellent 5mC methylation prediction on HX1. For example, on the chromosome 1, the AP value of DeepMod is 0.998 and the AUC value is 0.979, demonstrating highly accurate predictions. We note that nanopolish was designed for those CpG sites which were completely methylated/un-methylated CpG sites within 10 bps and not semi-methylated, while DeepMod does not have this limitation.

**Cross-species independent evaluation of 5mC prediction on E. coli.** To further evaluate cross-species performance of DeepMod on the E. coli data, we trained a model using chromosome 1–10 of NA12878 (where both NA12878 and E. coli data were basecalled with the same version of Albacore). We then used DeepMod to make methylation prediction on several E. coli data sets, including Con1, Con2, GCGC_HhaI, CG_MpeI, and CG_SssI[3]. For each of three positive data set (GCGC_HhaI or CG_MpeI or CG_SssI), we mixed motif-based 5mC with the corresponding motif sites in both Con1 and Con2 for performance evaluation of DeepMod, and the number of the cytosines of interest in motifs from positive control and from Con1 and Con2 were shown in Table 2. With a coverage ≥ 1, DeepMod achieved AP = 0.95 on GCGC_HhaI, AP = 0.807 on CG_MpeI, and AP = 0.973 on CG_SssI. The AUC values achieved by DeepMod are 0.942, 0.809, and 0.965 on GCGC_HhaI, CG_MpeI, and CG_SssI, respectively. On GCGC_HhaI and CG_SssI, the AP values were 2–4% points lower than the prediction by DeepMod trained on the E. coli data of CG_MSssI and UMR[4], whereas the AP value (0.807) was 0.11 lower on CG_MpeI. In summary, the cross-species testing suggested that DeepMod trained on one species can make accurate prediction on the other species, when the same basecalling algorithm was used. We note that the data from NA12878 was generated from native DNA while the data from Stoiber et al.[3] were treated by PCR and enzymes: native DNAs may contain incompletely methylated cytosines due to cellular heterogeneity, which may result in slightly lower performance when training a model.

**Performance of 6mA prediction on Nanopore sequencing data on E. coli.** To evaluate DeepMod on 6mA modifications within specific sequence motifs, we used three positive control data sets by Nanopore sequencing of E. coli with synthetically introduced 6mA[3], where M.TaqI, M.EcoRI and M.dam were used separately to methylase 30,914 TCGA motifs (denoted tcgA_TaqI for short), 1290 GAATTC motifs (denoted gaAttc_EcoRI for short), and 38,240 GATC motifs (denoted gAtc_dam for short), respectively. With region-based independent validation strategy, we evaluated DeepMod for the genomic positions from 1,000,000 to 2,000,000, whereas reads mapped outside of this region was used for training. After predicting methylation percentage for bases of interest in this region, we calculated AP and AUC values by combining each of the three 6mA positive control with Con1 and Con2 in Fig. 3a for those adenines in motifs, and the number of 6mA in motifs and the corresponding non-methylated adenines were given in Table 2. Figure 3a showed that DeepMod achieved AP = 0.884 and AUC = 0.874 on gaAttc_EcoRI, AP = 0.858, and AUC = 0.857 on tcgA_TaqI, and AP = 0.913 and AUC = 0.903 on gAtc_dam with coverage ≥ 5, suggesting that DeepMod performed well for 6mA modification detections. In particular, given a threshold of predicted methylation percentage ≥ 0.1 for a genomic position of interest, DeepMod achieved precision = 1.0 and recall = 0.788 on gaAttc_EcoRI, precision = 0.859 and recall = 0.841 on tcgA_TaqI, and precision = 0.748 and recall = 0.929 on gAtc_dam. We also conducted cross-validation of the five different regions in E. coli reference genome (i.e., [0, 1000000], [1000000, 2000000], [2000000, 3000000], [3000000, 4000000], [4000000, 4700000]), and DeepMod achieved AP = 0.83 ± 0.037 on gaAttc_EcoRI, AP = 0.89 ± 0.073 on gAtc_dam, and AP = 0.81 ± 0.041 on tcgA_TaqI, showing varying but similar performance in different genomic regions.

We next evaluated the performance of our methods on all methylated adenines regardless of whether adenines are within specific sequence motifs or not. When we calculated AUC using the predictions for all adenines, the AUC values increased from

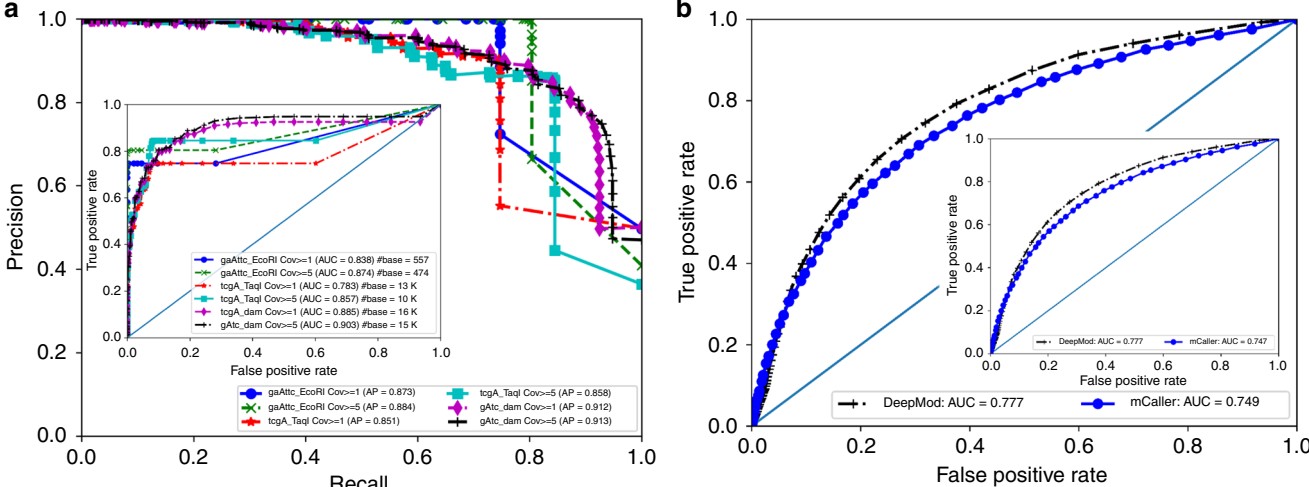

**Fig. 3** Evaluation of the performance of DeepMod on 6mA prediction on *E. coli* and *C. reinhardtii*. **a** AP (the outer) and AUC (the inner) plots on *E. coli* for three synthetically introduced 6mA data sets by EcoRI (gaAttc_EcoRI for GAATTC motif), TaqI (tcgA_TaqI for TCGA motif), and dam (gAtc_dam for GATC motif), respectively. **b** 6mA prediction by DeepMod on *C. reinhardtii* (the outer plot is for genomic DNA digested in 0.5 h, whereas the inner plot is for genomic DNA digested in 12 h[34]). Cov: coverage. # base: total number of bases used in the evaluation

0.874 to 0.898 for gaAttc_EcoRI, 0.857 to 0.92 for tcgA_TaqI, and 0.903 to 0.973 for gAtc_dam, indicating lower overall prediction errors. However, the AP values decreased. For example, for coverage ≥ 5, when we classified all adenines rather than gAtc-based adenines, the AP value decreased to 0.874, whereas the AP value decreased to 0.66 when we classified all adenines rather than tcgA-based adenines. This suggested a fraction of false positive predictions of 6mA not in motifs of interest, compared with a small number of true positive modifications. For example, there was only 277 6mA on gaAttc_EcoRI and 1,003,885 un-modified adenines, and even 1% misclassification of un-modified adenines is much larger than the number (277) of 6mA, and will lower the overall AP values.

**Cross-species independent evaluation of 6mA prediction on *C. reinhardtii*.** As *C. reinhardtii* is among the first eukaryotes[19,34,35] with detailed characterization of 6mA modifications, to further evaluate 6mA prediction of DeepMod, we sequenced *C. reinhardtii* using Nanopore sequencing with ~ 126X coverage, and then used the model trained on *E. coli* to make 6mA prediction for *C. reinhardtii*. The predictions were compared with the existing work of DA-6mA-seq results[34,35] where genomic DNA was digested in 0.5 h and 12 h. We plotted AUC for the DeepMod prediction together with the prediction by mCaller[26] with coverage ≥ 15 in Fig. 3b. mCaller used current deviations of six consecutive 6-mers around a position of interest as input of a neural network for 6mA prediction and it is known to be a highly accurate 6mA predictor. In this evaluation, mCaller achieved 0.74 AUC values for both DA-6mA-seq results, and DeepMod achieved slightly higher AUC value of 0.77 AUC. When we used coverage > 50 for both tools (since this data has ~ 126X coverage), DeepMod achieved AUC = ~ 0.8, whereas mCaller achieved AUC = ~ 0.75 for both DA-6mA-seq results. In summary, DeepMod can make accurate inferences on detection of 6mA modifications in cross-species evaluation studies.

## Discussion
DeepMod developed in this study bridges the gap between the rapid growth of Nanopore sequencing data and the increasing need of detecting DNA modifications at a genomic scale. DNA modifications are not consistent through biological cell cycle and

may vary between cell types. DNA modifications, such as 5mC, could be introduced by methyltransferases during cell development, and might also be de-methylated in some other biological processes such as cell differentiation. Specific enzymes are necessary to methylate cytosine or de-methylate 5mC, which plays a critical role in epigenetic reprogramming, whereas aberrant DNA modifications have been found to play important roles in human diseases such as various cancers[36]. Although bisulfite sequencing and PacBio long-read techniques have achieved great progress to detect some types of DNA modifications, especially 5mC, the advent of Nanopore long-read sequencing has significant potential to detect major types of DNA modifications from Nanopore raw electric signals. However, Nanopore sequencing data contains context-dependent electric signals which complicates the ability to pinpoint where modifications are located. Therefore, in this study, we propose a deep learning framework, DeepMod, to detect DNA modifications from raw electric signals of events (together with mapped base types) from Nanopore sequencing in a context-dependent manner via the use of LSTM. Evaluation on three types of species with two major type of DNA modifications demonstrates that DeepMod performs accurately to identify both types of DNA modifications and has great potential to be extended to other types of DNA modifications. DeepMod thus can be used as a genomic tool to detect genomic modifications for the increasing volume of Nanopore sequencing data without additional sequencing cost.

There are several current limitations in DeepMod that we wish to discuss here. First, DeepMod is trained and evaluated on 5mC and 6mA, two common DNA modifications, and it might not work well with other types of modifications or other different motifs before sufficient amounts of training data is available. Nevertheless, DeepMod provides a simple extension framework, and the model of other types of DNA modifications could be easily trained. This limitation could be further addressed when unsupervised or one-class learning are used to build a prediction model in the future, with a tradeoff that the specific modification types might not be predicted. Second, DeepMod can only work on DNA data of Nanopore sequencing currently. RNA data of direct Nanopore sequencing contains natural RNA modifications, which are informative to interpret gene expression and gene regulation, but large-scale ground-truth RNA modification data are currently lacking for supervised machine learning. In

addition, uracil in RNA may make Nanopore electric signals in RNA data different from that in DNA data, and thus additional work is needed to see whether the DeepMod framework needs modification to handle direct RNA Nanopore sequencing data. Third, DNA modifications could affect Nanopore signals of several neighboring bases, and the impacts from nearby modifications within a few bases away may influence the prediction of a given base, yet this fact is not currently considered in DeepMod. Fourth, DeepMod relied on alignment tool to find correct reference positions of bases in long reads. Now, DeepMod supports two widely used aligners: BWA-MEM and minimap2[32]. If different aligners or improper parameters are used with poor alignment performance, DeepMod's prediction will also be affected. Re-alignments with different tools or parameters might bring in new examined sites or exclude previously examined sites. This effect contributes much less to false modification prediction for those frequent modification types, but may become significant for those rarely occurring modifications. However, the majority of those limitations can be addressed by having more ground-truth modifications data sets for training the prediction models with proper parameters. When more and more modifications data sets become available, we believe that improved model in DeepMod could be learned to generate better prediction models.

In summary, DeepMod together with Oxford Nanopore sequencing provide a useful tool to identify DNA modifications at a genomic scale. We expect that DeepMod can be used in whole-genome epigenetics analysis with more types of modifications in future, with appropriately trained prediction models on additional gold standard data sets generated by the community.

## Methods

We developed DeepMod for the detection of DNA modifications from Nanopore sequencing data. DeepMod is a supervised deep RNN with Nanopore sequencing data as input. In this section, several Nanopore sequencing data sets used in this study were described below together with how to build, train and evaluate DeepMod on Nanopore sequencing data generated for *E. coli*, *C. reinhardtii,* and two human genomes (NA12878 and HX1).

**Existing Nanopore sequencing data**. Several Nanopore sequencing data sets were used in this study, including seven published positive and three negative control data sets for *E. coli* sequenced by Simpson et al.[4] and by Stoiber et al.[3], one published Nanopore sequencing data set for a human genome NA12878 sequenced by Jain et al.[31], a new Nanopore sequencing data set on HX1 and a new sequencing data set on *C. reinhardtii* both generated in this study.

The positive control samples for *E. coli* contain four data sets for 5mC and three data sets for 6mA where methylations were synthetically introduced by specific enzymes, whereas three negative control samples were amplified by PCR and thus contain no modified bases. A positive control of 5mC data set (with the short name CG_MsssI) and a negative control data set (with a short name of UMR) were sequenced by Simpson et al.[4], whereas the three 5mC data sets (with short names of GCGC_HhaI, CG_MpeI, and CG_SssI) and three 6mA data sets (with short names of tcgA_TaqI, gAtc_dam, and gaAttc_EcoRI) and two negative control data sets (with short names of Con1 and Con2) were sequenced by Stoiber et al.[3]. The detail description of these data sets can be found in Supplementary Notes and[3,4]. The sequencing data sets on two human genomes and on *C. reinhardtii* were based on native DNA molecules, and thus contain native modifications. Below, we will introduce Nanopore sequencing process for HX1 and *C. reinhardtii*, whereas the Nanopore sequencing data of NA12878 can be downloaded online.

**Nanopore sequencing data on HX1**. HX1 is a well-studied genome from a Chinese individual who has been sequenced by Illumina short-read sequencing and PacBio long-read sequencing[33]. Native human genomic DNA was extracted from fresh blood, and size selection was performed using Blue Pippin (Cassette kit: BUF7510; size range: 30–40 kb). DNA quality was assessed by running 1 μl on a genomic ScreenTape on the TapeStation 2200 (Agilent) to ensure a DNA Integrity Number > 7. Concentration of DNA was assessed using the dsDNA HS assay on a Qubit fluorometer (Thermo Fisher).

For library preparation, 2.0 μg size-selected (> 20 kb) genomic DNA was used as the input DNA of each library. End repair and dA-tailing was performed using NEBNext Ultra II End Repair/dA-tailing Module (catalog No. E7546). In all, 7 μl Ultra II End-Prep buffer, 3 μl Ultra II End-Prep enzyme mix were added to the input DNA. The total volume was adjusted to 60 μl by adding nuclease-free water (NFW). The mixture was incubated at 20 °C for 5 min and 65 °C for 5 min. A 1 ×

volume (60 μl) AMPure XP clean-up was performed and the DNA was eluted in 31 μl NFW. One microliter of the eluted dA-tailed DNA was quantified using the Qubit fluorometer. A total of ≥ 1.0 μg DNA should be retained if the process is successful.

Adaptor ligation was performed using the following steps. Twenty microliter Adaptor Mix (ONT, SQK-LSK108 Ligation Sequencing Kit) and 50 μl NEB Blunt/TA Master Mix (NEB, catalog No. M0367) were added to the 30 μl dA-tailed DNA. The mixture was incubated at room temperature for 10 min. The adaptor-ligated DNA was cleaned up using 40 μl of AMPure XP beads. The mixture of DNA and AMPure XP beads was incubated for 5 min at room temperature and the pellet was washed twice by 140 μl ABB (SQK-LSK108). The purified-ligated DNA was resuspended in 15.5 μl ELB (SQK-LSK108). A 1-μl aliquot was quantified by fluorometry (Qubit) to ensure ≥ 500 ng DNA was retained. The final library was prepared by mixing 35.0 μl RBF (SQK-LSK108), 25.5 μl LBB (SQK-LSK108), and 14.5 μl purified-ligated DNA. The library was loaded to R9.4 flow cells (FLO-MIN106, ONT) according to the manufacturer's guidelines. GridION sequencing was performed using default settings for the R9.4 flow cell and SQK-LSK108 library preparation kit. The sequencing was controlled and monitored using the MinKNOW software developed by ONT. Nanopore sequencing generated 4,827,155 FAST5 files in total after using Albacore v2.3.1 basecalling, and there were ~ 91G bases in total.

**Nanopore sequencing data on *C. reinhardtii***. *C. reinhardtii* has been used to study 6mA modification using DA-6mA-seq techinques[34] where a region containing 6mA modifications were detected. In the current study, we sequenced the same strain of *C. reinhardtii* as previously published, using Nanopore sequencing techniques as described above. We generated 772,817 FAST5 files after using Albacore v2.31 for basecalling, there were 15G bases in total, corresponding to ~ 126X genome-wide coverage.

**DeepMod framework**. DeepMod is a deep learning tool of bidirectional RNN with long short-term memory (LSTM) units. LSTM RNN is a class of artificial neural network for modeling sequential behaviors with LSTM to preclude vanishing gradient problem, and has achieved superior performance in handwriting recognition[37], speech recognition[38], and computational biology[39]. LSTM RNN would also be a better method to model series of raw signals generated in Nanopore sequencing.

In DeepMod, the input is a reference genome and FAST5 files generated by Nanopore sequencers with raw signals and base calls, and the output includes modification prediction for bases of interest in a long read and modification summary for genomic positions of interest in a reference genome in BED format. DeepMod contains several steps as shown in Fig. 1: (i) the alignment of long reads to a reference genome, (ii) feature extraction from inputs, (iii) a deep learning framework for modification prediction, and (iv) an optional second neural network for considering methylation cluster effect of 5mC in CpG sites, and (v) the modification summary for outputs. Each of the five steps was described below.

(i)   The input to DeepMod includes a reference genome and FAST5 files containing raw signals and events, which were generated by Nanopore sequencers with base calls. Each event is associated with a k-mer (e.g., 5-mer). In DeepMod, stay events without move were merged with the adjacent previous non-stay event with move to generate a single event. A sequence of bases associated with the resultant events was aligned to a reference genome using BWA-MEM with Nanopore-specific parameters[40] or minimap2. Each non-insertion base in long reads can be anchored with a genomic position in a reference genome for further analysis.

(ii)  In a FAST5 file, each event is associated with several sequential raw signals, and raw signals for all aligned bases in a long read were normalized using the method proposed by[3] and the normalized range was rescaled from −5 to 5. Then, the signal mean, standard deviation, and the number of signals associated with an event were extracted, plus a 4-vector feature for mapped reference bases. In the 4-vector feature, 1 indicates the mapped reference base is a specific nucleotide type, whereas 0 means otherwise. Thus, 7 features were used to describe an event (7-feature description for short and denoted by $\mathbf{x}_i = [f_m, f_d, f_b, f_A, f_C, f_G, f_T]$). In the training and testing process, events were also labeled by modification or non-modification inferred from synthetically introduced modifications for motifs of interest or from bisulfite sequencing (Please refer to the Supplementary Methods for more detail).

(iii) The deep learning framework in DeepMod is a bidirectional RNN of LSTM units as shown in Fig. 1. RNN is used to capture the Nanopore sequencing characteristic of that a signal would be affected by several neighborhood bases, and LSTM can overcome the effect of vanishing gradient problem in RNN training process[28]. LSTM RNN has been widely used in those fields where sequential order need to be preserved in training process of a deep learning framework such as handwriting recognition[37], speech recognition[38], and protein sequence analysis[41].

In DeepMod, bidirectional RNN with LSTM units was used with three hidden layers, where bidirectional RNN were used to take into consideration both forward and reverse data flow from neighborhood bases, and each LSTM unit contains multiple hidden nodes. In detail, an event and its $\lfloor w/2 \rfloor$

upstream and $\lfloor w/2 \rfloor$ downstream events (i.e., $\mathbf{x} = \{x_{i-\lfloor w/2 \rfloor}, \cdots x_i, \cdots x_{i+\lfloor w/2 \rfloor}\}$, where w is an odd number and $\lfloor w/2 \rfloor$ is the floor of $w/2$) were used as the inputs of $w$ LSTM units in RNN according to the sequential order in a long read, and 7 features ($x_i$) of each event was the input of a LSTM unit. In a LSTM unit with $x_j$ as input, $i-\lfloor w/2 \rfloor \leq j \leq i + \lfloor w/2 \rfloor$, then, the output $p_j$ is

$$p_j = \tanh(f^{i1} \circ f^{fi2} + v_{j-1} \circ f^f) \circ f^p \quad (1)$$

where $f^p = \text{sigmod}(f(\begin{matrix} a^p & b^p & c^p \end{matrix}))$ is activation function for output gate, $f^f = \text{sigmod}(f(\begin{matrix} a^f & b^f & c^f \end{matrix}))$ is activation function for forgot gate, $v_{j-1}$ is inner delay state, $f^{i1} = \tanh(f(\begin{matrix} a^{i1} & b^{i1} & c^{i1} \end{matrix}))$ is activation function for input, $f^{i2} = \text{sigmod}(f(\begin{matrix} a^{i2} & b^{i2} & c^{i2} \end{matrix}))$ is activation function for input gate, $f(\begin{matrix} a & b & c \end{matrix}) = [\begin{matrix} a & b & c \end{matrix}] * [\begin{matrix} x_j & p_{j-1} & 1 \end{matrix}]^T$, where superscript $T$ is the tranpose of a matrix, $p_{j-1}$ is the previous output, and $\circ$ is the element-wise multiplication.

Then, both forward and backward data flow were captured as shown in Fig. 1, and 3-layer RNN with full connections was used to capture complicated relationship between signals and prediction target of modifications. Given a long read with sequential events, an event and its neighborhood could be used as input of this neural network, and a prediction label was generated for this event. This process can be repeated one by one for events of interest in a long read and then for all long reads available. In the training process, the prediction labels $\hat{y}$ is treated by the softmax function, and cross-entropy $E$ would be minimized to tune the parameters in RNN.

$$E = y * -\log(\bar{y}) + (1-y) * -\log(1-\bar{y}) \quad (2)$$

where $y$ is the true labels of events regarding modifications, and $\bar{y} = \begin{bmatrix} \frac{e^{\hat{y}^n}}{e^{\hat{y}^n}+e^{\hat{y}^m}} & \frac{e^{\hat{y}^m}}{e^{\hat{y}^n}+e^{\hat{y}^m}} \end{bmatrix}$, and $\hat{y}^n$ is non-modification component for a prediction and $\hat{y}^m$ is modification component.

(iv) A second deep neural network was also designed for considering high correlation of 5mC modifications. Existing works[30] have demonstrated that 5mC modification in human genome are highly correlated with nearby CpG sites and with the corresponding sites at the opposite strand. We also investigated the methylation percentage of a CpG site and of its nearby CpG sites and the corresponding sites at the opposite strand in both replicates of bisulfite sequencing of NA12878, and found higher Pearson correlations between the methylation percentages of a CpG site and of its nearby CpG sites. To consider this effect, we designed the second deep neural network with four layers including an input layer, two hidden layers, and an output layer. The input of this layer is a 14-value vector: the predicted methylation percentage of a position, that of corresponding position of CpG site at the opposite strand, the number of nearby CpG sites within 25 bp, and the 11 discretized bins ([0, 0.05], [0.05, 0.15], [0.15, 0.25],…. [0.95, 1.0]) with value of how much percentage of nearby CpG sites has corresponding predicted methylation percentage. The first hidden layer has 100 hidden nodes, whereas the second hidden layer has 20 hidden nodes. Different layers were connected by full network with dropout (dropout rate = 0.7), and the output layer has sigmoid activation for outputting final methylation percentage for a CpG site. This network was only trained on the chromosome 1 of NA12878 with 100 epochs, and tested on all other chromosomes of NA12878 and HX1.

(v) In a real application, given a set of FAST5 files, DeepMod needs a reference genome to generate prediction modification for events of interest in a long read, and a sequence of bases from events were aligned with a reference genome. After the alignment, a genomic position of interest in a reference genome was aligned with a set of events each from a long read and with a prediction modification label. The modification information of genomic positions of interest was summarized in a BED format in a strand-specific manner with single-base resolution, and contained total coverage, modification coverage and modification percentage: if a position of interest was covered by a long read with forward strand alignment, its forward coverage was increased by 1, and further if the aligned base in the long read was predicted to be modified, its forward modification coverage was increased by 1; similarly, if a position of interest was covered by a long read with reverse strand alignment, its reverse coverage was increased by 1, and further if the aligned base in the long read was predicted to be modified, its reverse modification coverage was increased by 1. Modification percentage of genomic positions of interest was calculated using its modification coverage divided by its coverage in total. In a real-world application, there is no filter for prediction sites, and reads with mapped quality < 10 would not be used (only tens of reads among thousands of passed reads had poor mapped quality in our experiments.).

**Performance measurements.** For $M$ completely modified and $N$ completely un-modified bases (or motifs of interest), assume that DeepMod generated $P$ modified predictions and $Q$ un-modified predictions, accuracy, precision, recall, AP, and AUC are used to evaluate the performance.

$$\text{accuracy} = \frac{|P \cap M| + |Q \cap N|}{M + N} \quad (3)$$

$$\text{precision} = \frac{P \cap M}{P} \quad (4)$$

$$\text{recall} = \frac{|P \cap M|}{M} \quad (5)$$

$$\text{F1} - \text{score} = 2 * \frac{\text{precison} * \text{recall}}{\text{precision} + \text{recall}} \quad (6)$$

where $|*|$ is the size of $*$. Thus, accuracy is the fraction of correct predictions over all modified/un-modified cases in a data set, precision is a percentage of correct predictions of completely modified cases over all modified predictions, whereas recall is the number of correct prediction of completely modified cases divided by the number of completely modified cases in a data set. F1−score is a tradeoff metric of precision and recall. To evaluate binary classification of completely methylated positions and completely un-methylated positions in a reference genome, AP (which has been widely used in information retrieval) is weighted mean of precisions achieved at each threshold of predicted methylation percentage. As predicted methylation percentage decreases from 1 to 0, recall usually increases, whereas precision might decrease, and AP is in practice calculated by summing all precision with the recall difference at two adjacent thresholds in an ordered list as weights by scikit-learn. AP could evaluate how a classifier performs for the predictions of modifications (The prediction of un-modifications is not fully considered in this measurement). The range of the five measurements above is from 0.0 to 1.0. AUC is area under receiver operating characteristic curve, usually ranging from 0.5 to 1.0, and AUC can evaluate how a classifier performs for all predictions (considering both modifications and un-modifications). For all the measurements, the larger the value, the better the classification is.

**Training DeepMod.** We trained a 5mC model of DeepMod on the positive control CG_MSssI and the negative control UMR of E. coli[4]. To validate the deep learning model and select optimal hyper-parameters, two independent validation strategies were used on CG_MSssI and UMR. One strategy is read-based, where all long reads were divided into two groups: one group was used to train DeepMod with 90% positive-control long reads and 90% negative-control long reads, and the other group was used for testing DeepMod with 10% positive-control long reads and 10% negative-control long reads. Since the data set has thousands of bases in each of thousands of long reads, each time, a small portion of training data of positive control and another small portion of training data of negative control were loaded to train DeepMod, and this process was repeated until negative control data were loaded multiple times. Then, DeepMod was tested on the testing group. In this strategy, long reads in testing groups might align with same reference sequences as some reads in training data. Thus, we also employed a second strategy, which is a region-based independent validation, where all long reads or bases of long reads mapped to the genomic positions from 1,000,000 to 2,000,000 of E. coli were used for testing, no matter whether the long reads were from positive control or negative control; the rest of long reads were used for training DeepMod. The loading process for training DeepMod was similar to read-based independent validation strategy did above.

As no prior knowledge can guarantee which hyper-parameters would be optimal, to build a well-trained model for modification prediction, different hyper-parameters were tested during this training and validation process, such as window size $w$ from 7, to 11, to 15, to 21, to 31, and then to 51 with a step of 10. The detail performance of this validation was shown in Supplementary Table 1. According to the detail performance, the final well-trained model was selected for making methylation prediction on other data sets for 5mC prediction. A 6mA model of DeepMod was trained in a similar way on Con1, Con2, tcgA_TaqI, gAtc_dam, and gaAttc_EcoRI.

**Testing 5mC model of DeepMod on an independent E. coli data set.** 5mC and negative control Nanopore sequencing data sets from another research group[3] were used for cross-data set performance evaluation of DeepMod: the three 5mC data sets are GCGC_HhaI, CG_MpeI, and CG_SssI, whereas the negative control data sets are Con1 and Con2 (Please refer to the Supplementary Notes for their detail description). FAST5 files for each testing group (5mC data or negative control) together with a reference genome (E. coli strand K-12 sub-strand MG1655[42]) were taken as input of DeepMod, and each base of interest in long reads was predicted with a label to indicate whether it was methylated. Then, methylation prediction was summarized for each reference position of interest associated with a coverage, a methylation coverage and a methylation percentage. As in a positive control, a nucleotide of a specific motif were all methylated and negative control has no modification for all bases, we mixed each positive control with the negative control (Con1 and Con2), and calculated AP and AUC to evaluate the performance of DeepMod.

**Testing 5mC model of DeepMod on human data sets of NA12878 and HX1.** All long reads of NA12878 in FAST5 files together with a reference genome GRCh38 were used as input of DeepMod, and DeepMod made modification prediction for each base of interest in long reads with a label to indicate whether it was methylated. Then, predicted methylation was summarized for each genomic position of interest in GRCh38 associated with a total coverage, a methylation coverage and a methylation percentage. As native DNA sequencing in NA12878 contain naturally occurring modifications, and heterogeneity of sequenced DNA molecules generally exists, the criteria below was used to obtain high-quality methylation and non-methylation labels from bisulfite sequencing[32] of NA12878 for bases in long reads: for a cytosine of a position in GRCh38 with both > 90% of methylations and coverage $\geq c$ ($c$ would be 1, 3, 5, or 10) in two replicates of bisulfite sequencing, a cytosine in a long reads aligned with that position was considered to be modified and the cytosine at that position in the reference genome was called complete methylation; and if a cytosine of a position in GRCh38 has 0% methylations in both replicates of bisulfite sequencing, a cytosine in a long read aligned with that position was considered to be un-modified and the cytosine at that position in the reference genome was called complete un-methylation. Bases in long reads aligned with all other positions were not considered, and other positions of interest in GRCh38 without complete (un)-methylations were not used for binary classification performance. Then, the DeepMod prediction percentage of methylation at reference positions of interest (for example, CpG sites for 5mC) was compared against completely methylated or completely un-methylated reference positions, and AP and AUC were calculated to evaluate binary classification performance of DeepMod. The testing of DeepMod on HX1 data had the similar process to what we did for NA12878.

**Testing 6mA model of DeepMod on *C. reinhardtii*.** To test 6mA prediction on *C. reinhardtii*, we downloaded version 4.0 of *C. reinhardtii*[43] from JGI (Joint Genome Institute) as the reference genome, and input all FAST5 files of *C. reinhardtii* Nanopore sequencing data to DeepMod to obtain methylation prediction for each genomic position of interest in *C. reinhardtii*. The prediction results were compared with DA-6mA-seq results[34,35] for all GATC, TCGA. and GAATTCC motifs. In DA-6mA-seq, an adenine in motifs, which has an absolute score larger than 5 was considered as 6mA modifications, whereas other adenine in motifs is considered to not be modified. mCaller was run with a coverage $\geq 15$ and motifs of interest.

**Reporting summary.** Further information on research design is available in the Nature Research Reporting Summary linked to this article.

## Data availability

All Nanopore sequencing data on *C. reinhardtii* have been deposited to the EMBL-EBI European nucleotide archive under study PRJEB31789 [http://www.ebi.ac.uk/ena/data/view/PRJEB31789]. The raw human genome sequencing data sets on HX1 are available at http://hx1.wglab.org/. HX1 bisulfite sequencing data is available at the NCBI Sequence Read Archive (SRA) under the study PRJNA301527 [https://www.ncbi.nlm.nih.gov/bioproject/?term=PRJNA301527], while HX1 Nanopore sequencing data is available at the NCBI Sequence Read Archive (SRA) under the study PRJNA533926 [https://www.ncbi.nlm.nih.gov/bioproject/?term=PRJNA533926].

## Code availability

DeepMod is publicly available at https://github.com/WGLab/DeepMod and will be regularly maintained and updated. A detailed description of installing/running DeepMod and reproducible pipelines have also been documented in the GitHub repository.

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

## Acknowledgements

We thank Simpson et al. for making the Nanopore long-read data of *E. coli* publicly available and Jain et al. for making the Nanopore long-read data of NA12878 publicly available. We thank Dr. Guan-Zheng Luo (School of Life Sciences, Sun Yat-sen University) for providing the *C. reinhardtii* samples to us for the Oxford Nanopore sequencing. This study was supported by CHOP Research Institute to K.W. Nanopore sequencing of HX1 and *C. reinhardtii* genomes was supported by Guangdong Province Science & Technology Plan (2014B020228002) and National Natural Science Foundation of China (81530028).

## Author contributions

Q.L., C.X., and K.W. designed the study. Q.L. implemented the tool and performed the analysis. L.F. performed mCaller analysis of sequencing data. G.Y. and D.W. generated sequencing data for *C. reinhardtii* and HX1. C.X. prepared DNA samples for *C. reinhardtii* and HX1, guided on method development, and data analysis. Q.L. drafted the manuscript. All authors read, revised, and approved the manuscript.

## Additional information

**Competing interests:** G.Y. and D.W. are employees, and K.W. was a consultant of Grandomics Biosciences. The remaining authors declare no competing interests.

