## [Peer Review File · Nature Communications]

Reviewers' comments:

Reviewer #1 (Remarks to the Author):

The manuscript presents an approach to obtain information on modified DNA bases from Nanopore data. Although, using the raw 'squiggle' data to identify 5mC and 6mA is not a novel idea on its own, as is also acknowledged by the authors in the introduction, the implementation using deep learning looks promising and its reliability would make it a useful addition to the repertoire of tools and methods for Nanopore data analysis.

The code provided on GitHub appears well enough documented with instructions for installation and running, although appears to have no comments whatsoever in the python scripts.

Despite the promising results, I find that several major and minor concerns need to be addressed before it can be published in Nature Communications.

Major:

-- The structure of the paper does not really support easy understanding. In particular the fact that the description of Deepmod is completely saved for the Methods section, while the Results section does contain a whole section on "DeepMod training", seems to be counter intuitive. The reference to Figure 1 is made in the intro, but left without proper discussion. Moreover, the second deep learning model, which takes into account the correlation between nearby cytosines, is poorly explained / motivated. Can this be included in Figure 1?

-- The manuscript contains many errors in English grammar/style, so that several parts became hard to interpret. A thorough language check is recommended.

-- The comparison to other approaches is very superficial and incomplete. Why is a new deep learning model proposed, and not an existing basecaller re-trained with a 'fifth' base? There have been several deep learning approaches proposed for 'regular' base calling, including those with similar architecture (i.e. LSTM-based). So, it should be made more clear what exactly the novelty of the work is and/or it should be more clearly demonstrated that DeepMod truly outperforms straightforward extensions of existing methods. In this regard, I find it strange that Line 229 mentions using 'mcaller' for a comparison, but there is no further explanation on this tool, background, settings used or reasons why. Including its results requires some additional information.

-- The authors use quite a range of performance measures (which I like). Yet, the most important one, the average precision, is poorly explained (across which thresholds was averaged?). Moreover, I would suggest also including precision at a certain recall value (e.g. a value which would be considered acceptable). This would give a much more intuitive comparison (and for an operating point of the classifier which makes sense in practice). In this regard, the AUC is less revealing, since it is greatly influenced by the size of the negative set. I.e. I find it counterintuitive that the result (AUC) improves when the predictions for all adenines is used (line 215). It would be more convincing to report the AP here (or even better the precision at a certain recall).

-- The authors experiment with several combinations of training and testing data (i.e. random subset of reads, non-overlapping genome segments, different organism, in a specific motif or all cytosines, etc). It becomes a bit confusing to disentangle all the results, and I would recommend improving the presentation of the results to guide the reader better in the experimental settings that were used. I.e. Line 148-149: "...suggesting that the misclassification percentages have no significant decrease when all cytosines were considered", as opposed to only cytosines with what property exactly? GCGC vs CG

only?

-- Line 197-199: "We note that nanopolish was designed for those CpG sites which were completely methylated/un-methylated CpG sites within 10bps and not semi-methylated, while DeepMod does not have this limitation." How useful is this property as lines 152-155 seem to state that these situations are much less reliable?

-- Line 112-114: "In parallel, in the second region-based independent validation strategy, reads mapped to the genomic positions from 1,000,000 to 2,000,000 of E. coli reference genome were used for testing, whereas reads mapped to other genomic regions were used for training." Is there a particular reason this specific region was chosen? Would results change if this region would change (i.e. multiple folds in training and testing)?

-- Line 129-133: It is unclear why specifically the 3 used methylases were chosen. Is there a particular reason to target GCGC instead of CG, and is there a specific reason for these motifs? Are these all possible methylation sites known, and would the model need to be retrained for methylations that do not match these motifs? Why use 2 methylases for which the target motifs are the same?

-- Line 186-187: "To further evaluate DeepMod on an independent human blood sample HX1, we trained a model using completely methylated and completely un-methylated cytosines of chromosome 1 to 10 of NA12878".

Why these chromosomes specifically, and why retrained again?

Minor:

-- As mentioned above, the manuscript contains many smaller mistakes / grammar issues:

* "A 6mA model of DeepMod were trained"

* Modification summary (line 402 onward) is written in terms of 'could then would' rather than something that is actually implemented and tested.

* Line 420: "accuarcy"

* Several errors when referencing something in the manuscript: "Error! Reference source not found.", lines 340, 363, 380.

* Figure 3: "(a) 6mC predictions on E. coli". Probably should be 6mA instead.

* Supplementary data: Line 121, 123: "Error! Reference source not found."

-- Lines 166-170: "Furthermore, to account for the high correlations of nearby 5mC in CpG sites, we designed another deep neural network, and the second neural network incorporates the predicted methylation percentages of a CpG site, the prediction on the reverse strand and the methylation prediction of its neighbor bases (see Methods section), and outputs a final prediction percentage for a genomic position."

It is unclear how well this works based on just the priors and how much DeepMod actually adds in comparison at that point (i.e. we could just call the the most likely state of methylation instead). Additionally, the statement "on the reverse strand" is ambiguous, is this specific for 3'-5' vs 5'-3' or just 'the strand opposite of the one we are investigating'?

-- Line 94 does not completely match the approach described later in the paper: "DeepMod takes fast5 files generated from Nanopore sequencer as inputs, and extracts signal information of each event to determine whether modifications occur." Later on (line 345) the authors mention taking basecalled data, merge events where needed, and feed the new stats per basepair to the deep learner. This fact seems important to mention here as a deep learner on actual raw data directly would be significantly different.

-- Line 244-245: "Therefore, in this study, we propose a deep learning framework, DeepMod, to detect DNA modifications with Nanopore raw signals as input." While the tool does use raw data, leaving out the fact segmentation/basecalled information is also used oversells the tool here.

Line 345-346 states this as well: "The input to DeepMod are raw signals and events in a FAST5 file generated by Nanopore sequencers with base calls."

-- Line 248-249: "DeepMod thus could be used as a genomic tool to interpret genomic modifications for increasing volume of Nanopore sequencing data without additional sequencing cost." It seems this tool is useful to detect modifications rather than interpret.

Supplementary data:

-- Line 75-77: "Then, given a motif, for example, CpG sites, C in a CpG site was labelled to be methylated if (i) there were not more than 2 gaps in a 7-base window centered at that C, or (ii) there were not more than 3 gaps in a 13-base window centered at that C."

I'm unable to interpret what is happening here. What is a gap in this situation? And why would having a gap determine the methylation status?

The numbers seem rather arbitrarily chosen, also why first filter for 2 out of 7 and then provide a second chance with 3 out of 13? Is there any reasoning or statistics on this?

-- Line 94-98: "...on one hand, if a cytosine in a reference genome has both >90% of methylations in two replicates of bisulfite sequencing, all cytosines in long reads aligned with that position were considered to be completely methylated; on other hand, if a nucleotide in hg38 has 0% methylations in both replicates of bisulfite sequencing, all cytosine in long read aligned with that position were considered to be completely un-methylated."

The way it is currently written makes little sense, I assume this simply needs to be rewritten. As it currently stands, I interpret it as "if we find 100% evidence a particular C is methylated or not, we assume all C's in any read overlapping that position to be the same state", basically ignoring any information on other C's on the same reads. However, I assume what is meant is "if we find 100% evidence for the methylation state, this particular C on this genomic position is assumed to be the same state in any long read we sequenced".

-- Line 98-99: Nucleotides with methylation percent between 0% and 90% were not considered in this testing process.

There is no further information on how many positions are kept, removed, why these thresholds, and what happens if they shift one way or another.

Reviewer #2 (Remarks to the Author):

The authors described a novel deep learning-based model for methylation calling from nanopore sequencing data, and benchmarked it's power for accurately calling 5mC and 6mA. This is a really impressive study and definitely should be published on Nature communications. However I do have few minor concerns that need to be addressed by the authors.

Modification Importance:

Give a brief literature review on the importance of methylation in biological processes and the known kinds of methylation within DNA. Expand the importance of the methylation enzymes within E.coli and

human cell regulation.

Line 45-46 "DNA base modifications widely exist in different organisms and are essential in various biological processes"

Line 235-239 "DNA modifications are a special type of DNA variations without changing nucleotide types and plays a crucial role to control DNA translations and replications. Aberrant DNA modifications have been identified to play important roles in human diseases such as cancers. Genomic detection of DNA modifications could serve to improve the fundamental understanding of how epigenetics connect with cancer development."

Speak on DeepSignal comparison to DeepMod

DeepSignal and DeepMod both use similar data inputs and network architectures to accomplish the same binary classification task. Since DeepSignal analyzed the same NA12878 CpG data so there should be at a note about a comparison in performance. Completing a more in-depth model architecture and performance comparison with DeepSignal would be great but DeepSignal is not open source right now so it is not feasible to do a complete comparison.

Make Input to Network Clear

There are multiple points in the paper which make it seem as if the alignment information is not used as input to the network.

These sound like no alignment information is passed into the network.

Line 94 - 96 : "DeepMod takes fast5 files generated from Nanopore sequencer as inputs, and extracts signal information of each event to determine whether modifications occur. DeepMod also summarizes whole-genome modification information after aligning long reads in fast5 files to a reference genome."

Line 337-339 "In DeepMod, the input is FAST5 files generated by Nanopore sequencers with raw signals and base calls, and the output include modification prediction for each base in a long read and modification summary for each genomic position in a reference genome in BED format"

Line 345-346 "The input to DeepMod are raw signals and events in a FAST5 file generated by Nanopore sequencers with base calls"

Line 403-406 "In a real application, given a set of FAST5 files, DeepMod can generate prediction modification for each event in a long read, and a sequence of bases from events could be aligned with a reference genome. Then, a genomic position in a reference genome would be aligned with a set of events each from a long read and with a prediction modification label."

This make it seem like the mapping information is passed into the network

Line 354-355 "Then, the signal mean, standard deviation, and the number of signals 355 associated with an event were extracted, plus a 4-vector feature for mapped reference bases."

Input is not clear until this sentence.

Line 354-358 "Then, the signal mean, standard deviation, and the number of signals associated with an event were extracted, plus a 4-vector feature for mapped reference bases. In the 4-vector feature, 1 indicates the mapped reference base is a specific nucleotide type, while 0 mean otherwise. Thus, 7 features were used to describe an event (7-feature description for short and denoted by $x_i = [f_m, f_d, f_l, f_a, f_c, f_g, f_t]$."

Make more calls on NA12878:

Try making calls on less confident CpG sites and see if proportion of calls matches the proportion of modification in the bisulfite sequencing data.

Make note that evaluation of R9.4 data with model trained from R9 data.

It sounds like a model trained with the E.coli R9 chemistry was able to predict accurately the NA12878 sequencing data from R9.4. If this is the case then that is worth speaking about. If this is the case it may be interesting to take the R9.4 model trained using NA12878 and make inference calls on the E.coli data.

Supp Line 11-12 "This Nanopore sequencing dataset contains both positive control data and negative control data of E. coli. Both were sequenced by Simpson et. al. using Nanopore R9 sequencing techniques"

Line 182-184 "DeepMod 183 had the AP value equal to 0.998 and the AUC value to 0.995, indicating accurate prediction on human 184 genome, even when using a DeepMod model trained on E. coli data"

Confused between, per-read, per-reference-site and per-call performance.

You seem to change the performance measurement between per-reference-site, per-read and per-call accuracy. It seems like the E.coli accuracy is measuring per-read methylation accuracy whereas the NA12878 accuracy is measuring aggregated per-reference-site accuracy.

In the supplementary tables you report per-read performance but I would want to see the performance on every called base. The classifier is only making one per-site prediction so aggregating calls into a per-read performance metric blurs the true accuracy of the classifier. Per-read accuracy depends on the length of the read and the number of called sites within that read.

-- Per read

Supp Line 119-121 - "To calculate per-read performance, each read would be evaluated individually with precision, recall, accuracy and F1-score on CG_MSssl and with accuracy on UMR"

-- Per-reference-site

Line 162-164 "We made 5mC prediction for each Nanopore long read, and each cytosine in a long read was associated with a prediction label indicating whether it was predicted to be modified or not. Then, after all long reads were aligned against a reference genome, predicted methylation summary for the whole genome was generated for each cytosine in the genome with a percentage to indicate how many modification predictions were made for that position, compared to all reads covering that position."

-- per-cal

Line 177-179: "We compared DeepMod prediction against the completely methylated and unmethylated bases for each chromosome"

E.coli was done per-genomic position?

Line 463-465: "Then, methylation prediction was summarized for each genomic position associated with a coverage, a methylation coverage and a methylation percentage."

NA12878 evaluation was done per call?

Lin 480-483 "Then, the DeepMod prediction was compared with this definition, and AP and AUC were calculated to evaluate binary classification performance of DeepMod."

Be specific about training, testing and validation numbers:

It is difficult to find out how many reads and individual sites/motifs were used for training for each model. Specifically how many sites were used for training, validation and test?

How long did it take to train?

How many iterations?

How many times did the model see each individual site?

What compute resources were used?

How long does it take to do inference on X number of reads on X number of GPU's/CPU's?

What were your filtering steps for reads and for sites?

How many reads and sites were lost from each filter step?

Line 468-471: "All long reads of NA12878 in FAST5 files together with a reference genome GRCh38 were used as input of DeepMod, and DeepMod made modification prediction for each bases in long reads with a label to indicate whether it was methylated."

Limitations of Software:

Should mention how the software is not just limited by the type of modification but also by the motif that the modification is found. Also, species-specificity is a concern. Whether there're some species-specific methylation contexts?

Line 250-252: "There are several current limitations in DeepMod that we wish to discuss here. First, DeepMod is trained and evaluated with 5mC and 6mA, two commonly available DNA modifications, and thus it might not work well with other types of modifications."

Also, once you note that nanopolish has a limitation in CpG sites, you should explain your limitation within the main paper. If the selection criteria for sites depends on the CIGAR string (which can change based on which aligner is used), it should be noted that re-alignments can bring in or exclude sites previously examined.

Line 197-199: "We note that nanopolish was designed for those CpG sites which were completely methylated/un-methylated CpG sites within 10bps and not semi-methylated, while DeepMod does not have this limitation"

Supp Line 76-78 "Then, given a motif, for example, CpG sites, C in a CpG site was labelled to be methylated if (i) there were not more than 2 gaps in a 7-base window centered at that C, or (ii) there were not more than 3 gaps in a 13-base window centered at that C."

Error! Reference source not found:

There are several places where the reference to a table or citation is not found.

I only see 13 values of the 14 value vector for second deep neural network:

- 1: the predicted methylation percentage of a position
- 2: corresponding position of CpG site at the reverse strand
- 3: the number of nearby CpG sites within 25bp
- 4-13: the 10 discretized bins

Line 393-396: " The input of this layer is a 14-value vector: the predicted methylation percentage of a position, that of corresponding position of CpG site at the reverse strand, the number of nearby CpG sites within 25bp, and the 10 discretized bins ([0, 0.05], [0.05, 0.15], [0.15, 0.25],.... [0.95, 1.0]) with value of how many percentage of nearby CpG sites has corresponding predicted methylation percentage. "

Point-by-point authors' response

Summary of changes

We thank the reviewers and editors for your helpful evaluation of our manuscript titled “Detection of DNA base modifications by deep recurrent neural network on Oxford Nanopore sequencing data”. We have addressed all comments raised by the two Reviewers. As requested by the editor, major changes were also yellow-highlighted in the revised manuscript. The point-by-point responses to the reviewers' specific comments follow:

Reviewers' comments:

Reviewer #1 (Remarks to the Author):

The manuscript presents an approach to obtain information on modified DNA bases from Nanopore data. Although, using the raw ‘squiggle’ data to identify 5mC and 6mA is not a novel idea on it's own, as is also acknowledged by the authors in the introduction, the implementation using deep learning looks promising and its reliability would make it a useful addition to the repertoire of tools and methods for Nanopore data analysis.

The code provided on GitHub appears well enough documented with instructions for installation and running, although appears to have no comments whatsoever in the python scripts.

Response: Thank you very much for your nice summary of the manuscript, and for your helpful comments regarding GitHub code. To address these comments, we have now substantially improved the Python scripts to add more comments, to help users understand and modify the code. The updated GitHub repository is at <https://github.com/WGLab/DeepMod>.

Despite the promising results, I find that several major and minor concerns need to be addressed before it can be published in Nature Communications.

Response: Thank you for your suggestions. We have addressed all your concerns as detailed below.

Major:

-- The structure of the paper does not really support easy understanding. In particular the fact that the description of Deepmod is completely saved for the Methods section, while the Results section does contain a whole section on “DeepMod training”, seems to be counter intuitive. The reference to Figure 1 is made in the intro, but left without proper discussion. Moreover, the second deep learning model, which takes into account the correlation between nearby cytosines, is poorly explained / motivated. Can this be included in Figure 1?

Response: Thank you very much for this suggestion. We have now added a Method Summary just before the description of Results and discussed more about the method (including the second neural network which has been included in Figure 1). See pages 4-5 (1st paragraph). Additionally, the second neural network is motivated by (1) Lovkvist et al NAR 2016 whose conclusion is that methylation of CpG sites in CpG islands is not independent but affected by methylation of surrounding sites, and (2) our investigation of the methylation percentage of a CpG site and of its nearby CpG sites and the corresponding sites at the opposite strand in both replicates of bisulfite sequencing of NA12878. We found higher Pearson correlations between the methylation percentages of a CpG site and of its nearby CpG sites. We added these motivations in the design of the second neural network (see page 16 paragraph 2). We also found that the second deep neural network could improve AP by 1-3 percent points and AUC by 3-5 percent points for all chromosomes in NA12878 when coverage ≥ 1 (see page 7 last paragraph).

-- The manuscript contains many errors in English grammar/style, so that several parts became hard to interpret. A thorough language check is recommended.

Response: Thank you. We have carefully checked the manuscripts and corrected errors, with the assistance of a colleague.

-- The comparison to other approaches is very superficial and incomplete. Why is a new deep learning model proposed, and not an existing basecaller re-trained with a 'fifth' base? There have been several deep learning approaches proposed for 'regular' base calling, including those with similar architecture (i.e. LSTM-based). So, it should be made more clear what exactly the novelty of the work is and/or it should be more clearly demonstrated that DeepMod truly outperforms straightforward extensions of existing methods. In this regard, I find it strange that Line 229 mentions using 'mcaller' for a comparison, but there is no further explanation on this tool, background, settings used or reasons why. Including its results requires some additional information.

Response: Thank you very much for the suggestion, and we would like to take this opportunity to explain the rationale. Extending an existing basecaller with a 'fifth' base may be feasible but it is very challenging now due to several factors below: (1) Currently all standard base calling (Albacore or Guppy) is performed by the manufacturer and it has been well developed. Only binary is available and in the case of Guppy, the basecalling tool is already integrated into sequencer itself, so it is not possible for us to modify the code to call the fifth base; (2) a 'fifth' base of modification has very similar signal distribution as the unmodified bases (since only a methyl group is added), yet different bases can differ substantially due to large structural differences of nucleotides. Given the context-dependency of signals at each base position, it is very unlikely for a machine-learning approach to perform well when classifying five or six types, when four of them differ substantially and all of them have context dependency; (3) as mentioned above, the modifications not only affect the signals of modified positions but also the signals of nearby positions with different base types, which makes the signal distribution more complicated and may require different types of models than traditional base caller; (4) given limited signal range, increasing a 'fifth' base would increase the possible number of 5-mer from $4^5=1025$ to $5^5=3125$, which would result in more error rates in basecalling given that the error rate of 1025 5-mers is already higher; (5) the majority of modified bases can be called with

the correct base types, suggesting that the signal difference of modified bases and un-modified bases might be minimal compared to the signal difference between different base types. Thus, extending an existing basecaller Guppy with a ‘fifth’ base is not an ideal option at the moment.

Additionally, according to your suggestion, we also make the description of mcaller clearer. This tool was in fact described in the introduction (ref. 26) and was developed by McIntyre et al, but in the previous version of the manuscript, the mcaller name was not associated with the discussion of this work. In the revised version, we have clearly related this work with mcaller, and also discussed more about mcaller in the comparison (page 4 paragraph 1, page 11 paragraph 1 and page 19 last paragraph).

-- The authors use quite a range of performance measures (which I like). Yet, the most important one, the average precision, is poorly explained (across which thresholds was averaged?). Moreover, I would suggest also including precision at a certain recall value (e.g. a value which would be considered acceptable). This would give a much more intuitive comparison (and for an operating point of the classifier which makes sense in practice). In this regard, the AUC is less revealing, since it is greatly influenced by the size of the negative set. I.e. I find it counterintuitive that the result (AUC) improves when the predictions for all adenines is used (line 215). It would be more convincing to report the AP here (or even better the precision at a certain recall).

Response: Thank you for these comments. We added more explanation of average precision (AP): “To evaluate binary classification of completely methylated positions and completely unmethylated positions in a reference genome, AP (which has been widely used in information retrieval) is weighted mean of precisions achieved at each threshold of predicted methylation percentage. As predicted methylation percentage decreases from 1 to 0, recall usually increases while precision might decrease, and AP is in practice calculated by summing all precision with the recall difference at two adjacent thresholds in an ordered list as weights. AP could evaluate how a classifier performs for the predictions of modifications”. (See page 17 paragraph 3)

According to your suggestion, we also added precision and recall for classifying completely methylated and unmethylated motif-based cytosines (adenines) on two human genomes (Supplementary Table 4 and 5) and on E. coli data when prediction methylation percentage is larger than a threshold. For example, for evaluating motif-based 5mC, given a threshold of prediction methylation percentage ≥ 0.1 for a genomic position of interest, DeepMod gave precision=0.945 and recall=0.97 for GCGC_HhaI, precision=0.835 and recall=0.879 for CG_MpeI and precision=0.848 and recall=0.986 for CG_SssI. DeepMod achieved precision=0.96 and recall=0.985 when the threshold is set to 0.2. (See pages 6-7) Further for evaluating motif-based 6mA, given a threshold of predicted methylation percentage ≥ 0.1 for a genomic position of interest, DeepMod achieved precision=1.0 and recall=0.788 on gaAttc_EcoRI, precision=0.859 and recall=0.841 on tcgA_TaqI, and precision=0.748 and recall=0.929 on gAtc_dam. (See page 10 paragraph 1)

Regarding your concern in line 215, we included both AP values and AUC values. We also agree with you that when class ratio of positive and negative set is very unbalanced, and that AP is

better than AUC for the performance evaluation. (See page 10 paragraph 2)

-- *The authors experiment with several combinations of training and testing data (i.e. random subset of reads, non-overlapping genome segments, different organism, in a specific motif or all cytosines, etc). It becomes a bit confusing to disentangle all the results, and I would recommend improving the presentation of the results to guide the reader better in the experimental settings that were used. I.e. Line 148-149: "...suggesting that the misclassification percentages have no significant decrease when all cytosines were considered", as opposed to only cytosines with what property exactly? GCGC vs CG only?*

Response: Thank you for these suggestions and comments. We divided the results first according to methylation types (5mC or 6mA). Then, for 5mC (or 6mA) predictions, our testing results were organized as you suggested: on random subset of reads, on non-overlapping genome, cross-datasets, cross-organism. Based on your suggestion, we further clarified the procedure on motif-based evaluation and all cytosines (adenines) evaluation on E. coli data.

Further, to remove the confusion in line 148-149, we rephrased the sentences as “DeepMod achieved AUC of 0.985, 0.953 and 0.992 for all cytosines in the GCGC_Hhal dataset, the CG_MpeI dataset and the CG_SssI dataset respectively, suggesting that the misclassification percentages have no significant decrease when moving the classification of motif(GCGC or CG)-based cytosines to the classification of all cytosines. However, from Figure 2(c), the AP values of DeepMod to classify all cytosines are 0.771, 0.890 and 0.983 in the GCGC_Hhal dataset, the CG_MpeI dataset and the CG_SssI dataset respectively, which were smaller than the performance for the prediction of cytosines in motifs, indicating misclassifications for cytosines that are not present in motifs.” (Page 7 paragraph 2)

-- *Line 197-199: "We note that nanopolish was designed for those CpG sites which were completely methylated/un-methylated CpG sites within 10bps and not semi-methylated, while DeepMod does not have this limitation." How useful is this property as lines 152-155 seem to state that these situations are much less reliable?*

Response: The two groups of sentences (Lines 197-199 and Lines 152-155) discussed two individual cases. In Lines 197-199, the assumption of nanopolish is that all 5mC methylation in CpG sites within 10bps and CpG sites of the opposite strand have the same methylation states. This assumption might be useful for some cases, but might not be true for other cases especially non-5mC modifications.

In contrast, lines 152-153 discussed those C not in CpG sites which was not investigated in nanopolish, and lines 152-153 discussed the neighborhood effect of a modification---a modification might change Nanopore signals at both the modified position and also its neighborhood positions (especially adjacent positions). For example, the jth position has a modification and its Nanopore signals differ from its unmodified signals, and at the same time, the j+1th/j-1th position might has different Nanopore signals from its unmodified cases. This is the effect of the modification at jth position (this is the property of Nanopore signals). The signal change of the j+1th position is a redundant evidence that there is a modification at the jth

position which is correct. But in the evaluation, we considered the modification prediction at jth position to be correct and the redundant to be incorrect, and thus this kind of prediction evaluation indicates the underestimation of the prediction power of our method.

-- Line 112-114: *"In parallel, in the second region-based independent validation strategy, reads mapped to the genomic positions from 1,000,000 to 2,000,000 of E. coli reference genome were used for testing, whereas reads mapped to other genomic regions were used for training." Is there a particular reason this specific region was chosen? Would results change if this region would change (i.e. multiple folds in training and testing)?*

Response: This region was arbitrarily chosen in our original evaluation. According to your suggestions, we also selected regions individually from 0 to 1,000,000, from 2,000,000 to 3,000,000, from 3,000,000 to 4,000,000 and from 4,000,000 to 4,700,000 for testing with corresponding different trained models, and does not see the significant deviations of the performance in different regions (see Supplementary Table 2). We also conducted similar multiple-fold testing for 6mA prediction, and there is also insignificant deviation of performance in different regions. The updated results are shown in page 6 paragraph 1 and page 10 paragraph 1.

-- Line 129-133: *It is unclear why specifically the 3 used methylases were chosen. Is there a particular reason to target GCGC instead of CG, and is there a specific reason for these motifs? Are these all possible methylation sites known, and would the model need to be retrained for methylations that do not match these motifs? Why use 2 methylases for which the target motifs are the same?*

Response: The 3 methylase Nanopore datasets of 5mC were sequenced by Stoiber et. al. bioRxiv 2017 (We sequenced a human genome HX1 and an algae genome). These are specific methylases that target sequence motifs such as CG and GCGC so that "gold standard" can be introduced to genome to generate a 5mC benchmarking data set for analysis. Two methylases were used for a same target motif because they are two widely-used methylases for introducing 5mC. Since our model uses supervised learning, the motif information in training data may result in over-fitting so that the model does not work well with other motifs; therefore, it is important to train and test on different data sets with defined motifs (such as the methylase introduced methylations) or without defined motifs (such as naturally occurring methylations in human blood or cell lines) as described in the paper.

-- Line 186-187: *"To further evaluate DeepMod on an independent human blood sample HX1, we trained a model using completely methylated and completely un-methylated cytosines of chromosome 1 to 10 of NA12878". Why these chromosomes specifically, and why retrained again?*

Response: There is no other specific reason: There are 23 pairs of chromosomes in human genome and we picked the first 10 chromosomes for training process in deep learning process (as shown in Supplementary Table 4). By testing on different sets of chromosomes, we address one question raised in the comment above, that is, the reliance of specific sequence contexts/motifs in

prediction model. Since sequences in chromosome 1-10 differ from the other chromosomes, this analysis reduces concerns on overfitting on sequence contexts.

The reason why we trained the model again is to reduce the effect of different versions of Albacore on the event generation. Albacore is a standard basecalling method of Nanopore data designed by Nanopore Company. However, we found that the signals generated by different versions of Albacore are inconsistent, and a model trained on one version might generate non-optimal prediction on the other version. Another solution is to re-basecall all Nanopore data with a single version. However, Albacore has been quickly developed since 2016 from an older version of <v0.8.2 to a latest version of v2.3.4. The E coli Nanopore data (used in the first training model) were sequenced using older Nanopore flowcell and kits which are not supported since v0.8.2 and most of older version of Albacore is not available, while HX1 was sequenced with much newer Nanopore flowcell and kits. Thus, we only used the same version of Albacore for basecalling both NA12878 and HX1 to reduce the biases caused by different software versions. We thus re-trained a model for the prediction on HX1 to reduce this effect of different versions of Albacore.

Minor:

-- As mentioned above, the manuscript contains many smaller mistakes / grammar issues:

* "A 6mA model of DeepMod were trained"

* Modification summary (line 402 onward) is written in terms of 'could then would' rather than something that is actually implemented and tested.

* Line 420: "accuarcy"

* Several errors when referencing something in the manuscript: "Error! Reference source not found.", lines 340, 363, 380.

* Figure 3: "(a) 6mC predictions on E. coli". Probably should be 6mA instead.

* Supplementary data: Line 121, 123: "Error! Reference source not found."

Response: Thank you very much for your suggestions. We have corrected all the aforementioned issues and also checked all documents for other issues.

-- Lines 166-170: "Furthermore, to account for the high correlations of nearby 5mC in CpG sites, we designed another deep neural network, and the second neural network incorporates the predicted methylation percentages of a CpG site, the prediction on the reverse strand and the methylation prediction of its neighbor bases (see Methods section), and outputs a final prediction percentage for a genomic position."

It is unclear how well this works based on just the priors and how much DeepMod actually adds in comparison at that point (i.e. we could just call the the most likely state of methylation instead). Additionally, the statement "on the reverse strand" is ambiguous, is this specific for 3'-5' vs 5'-3' or just 'the strand opposite of the one we are investigating'?

Response: We corrected "the reverse strand" with "the opposite strand" according to your suggestions. We checked the improvement of the second neural network on NA12878 with coverage \geq 1, and found that AP across chromosomes was improved by 1-3 percent point while the AUC values increased by 3-5 percent point. (see page 7 last paragraph).

-- Line 94 does not completely match the approach described later in the paper: "DeepMod takes fast5 files generated from Nanopore sequencer as inputs, and extracts signal information of each event to determine whether modifications occur." Later on (line 345) the authors mention taking basecalled data, merge events where needed, and feed the new stats per basepair to the deep learner. This fact seems important to mention here as a deep learner on actual raw data directly would be significantly different.

-- Line 244-245: "Therefore, in this study, we propose a deep learning framework, DeepMod, to detect DNA modifications with Nanopore raw signals as input." While the tool does use raw data, leaving out the fact segmentation/basecalled information is also used oversells the tool here.

Line 345-346 states this as well: "The input to DeepMod are raw signals and events in a FAST5 file generated by Nanopore sequencers with base calls."

Response: Thank you very much for these comments to improve the description on the method. We have rewritten Line 94, Lines 244-245 and other lines to be consistent. For your quick reference, the input of deep learning model in DeepMod is the summary of raw signals of an event and mapped base type (A, C, G or T) of the event, while DeepMod takes a reference genome and fast5 files (with raw signals and event information) as input.

-- Line 248-249: "DeepMod thus could be used as a genomic tool to interpret genomic modifications for increasing volume of Nanopore sequencing data without additional sequencing cost." It seems this tool is useful to detect modifications rather than interpret.

Response: Thank you. We corrected "interpret" by using "detect".

Supplementary data:

-- Line 75-77: "Then, given a motif, for example, CpG sites, C in a CpG site was labelled to be methylated if (i) there were not more than 2 gaps in a 7-base window centered at that C, or (ii) there were not more than 3 gaps in a 13-base window centered at that C."

I'm unable to interpret what is happening here. What is a gap in this situation? And why would having a gap determine the methylation status?

The numbers seem rather arbitrarily chosen, also why first filter for 2 out of 7 and then provide a second chance with 3 out of 13? Is there any reasoning or statistics on this?

Response: Thank you for these comments. This is a process of obtaining reliable motif-based 5mC in training process only (motif-based 5mC were synthetically introduced by methylases). To do that, we first aligned long reads with a reference genome, and then according to the alignment, we detected 5mC in reliable alignments with less gaps. In other words, more gaps in a small region indicates poor local alignment and those motifs in such poor alignment regions were thus not used. This process is used to reduce wrong labeling of methylated genomic positions rather than determine methylation status. We added more description here to void this confusion.

7-based window (the minimum odd number larger than 5. 5-mer is usually used for event in fast5 files) and 13-base window (13 is almost two fold of 7) is based on our understanding of the alignment. The second window is used because methylation might result in more erroneous base

calls in neighborhood regions, and the reliable alignment in 13-base window indicates that poor alignment of 7-based window might be due to methylation effect rather than alignment errors. 7-based window was used first to reduce other effect on 13-base window. These numbers is selected in the model and may not be optimal.

-- Line 94-98: *"...on one hand, if a cytosine in a reference genome has both >90% of methylations in two replicates of bisulfite sequencing, all cytosines in long reads aligned with that position were considered to be completely methylated; on other hand, if a nucleotide in hg38 has 0% methylations in both replicates of bisulfite sequencing, all cytosine in long read aligned with that position were considered to be completely un-methylated."*

The way it is currently written makes little sense, I assume this simply needs to be rewritten. As it currently stands, I interpret it as "if we find 100% evidence a particular C is methylated or not, we assume all C's in any read overlapping that position to be the same state", basically ignoring any information on other C's on the same reads. However, I assume what is meant is "if we find 100% evidence for the methylation state, this particular C on this genomic position is assumed to be the same state in any long read we sequenced".

Response: Thank you for this comment. We have rephrased the sentences to avoid the confusion (page 19 paragraph 2 and page 3 paragraph 4 of supplementary). Your second interpretation is what we meant.

-- Line 98-99: *Nucleotides with methylation percent between 0% and 90% were not considered in this testing process.*

There is no further information on how many positions are kept, removed, why these thresholds, and what happens if they shift one way or another.

Response: The number of positions which were kept (completely methylated/unmethylated) and removed were provided in Supplementary Table 4 for NA12878. The thresholds were used to obtain these positions which might be complete methylation and complete un-methylation in two replicates in bisulfite sequencing due to the heterogeneity of sequencing data (sequenced by bisulfite sequencing or by Nanopore). Different thresholds might be used. For the threshold of complete methylation, a smaller threshold than 90% might introduce more positions with more methylation variability, while a larger threshold might reduce the number of methylation for evaluation. The threshold for complete un-methylation has the similar opposite scenarios. Different threshold might be used, and 0% and 90% is not optimal.

Reviewer #2 (Remarks to the Author):

The authors described a novel deep learning-based model for methylation calling from nanopore sequencing data, and benchmarked it's power for accurately calling 5mC and 6mA. This is a really impressive study and definitely should be published on Nature communications. However I

do have few minor concerns that need to be addressed by the authors.

Response: Thank you for your helpful comments. We have addressed all your concerns as described below.

Modification Importance:

Give a brief literature review on the importance of methylation in biological processes and the known kinds of methylation within DNA. Expand the importance of the methylation enzymes within E.coli and human cell regulation.

Line 45-46 “DNA base modifications widely exist in different organisms and are essential in various biological processes”

Line 235-239 “DNA modifications are a special type of DNA variations without changing nucleotide types and plays a crucial role to control DNA translations and replications. Aberrant DNA modifications have been identified to play important roles in human diseases such as cancers. Genomic detection of DNA modifications could serve to improve the fundamental understanding of how epigenetics connect with cancer development.”

Response: According to your suggestion, a brief overview of DNA methylation, their importance and known methylation types in DNA were discussed in the first paragraph (page 3 paragraph 1), and the importance of methylation enzymes was discussed in the Discussion section (page 11 paragraph 1).

Speak on DeepSignal comparison to DeepMod

DeepSignal and DeepMod both use similar data inputs and network architectures to accomplish the same binary classification task. Since DeepSignal analyzed the same NA12878 CpG data so there should be at a note about a comparison in performance. Completing a more in-depth model architecture and performance comparison with DeepSignal would be great but DeepSignal is not open source right now so it is not feasible to do a complete comparison.

Response: Thank you for your comment. We have added the follow sentence for a note of a direct comparison “We also note DeepSignal trained in E. coli tried to make cross-species prediction of CpG methylation on NA12878, which achieved F1=0.949 (with precision=0.97 and recall=0.93) on synthetically introduced 5mC in human while nanopolish achieved F1=0.89 (with precision=0.944 and recall=0.844). Our method on native 5mC prediction in NA12878 achieved F1=0.983 (with precision=0.988 and recall=0.979) and F1 values on each individual chromosome (calculated based on precision and recall in Supplementary Table 4) were about 0.98. Although this is not a direct comparison since DeepSignal is not open source right now, DeepMod still demonstrated its accurate prediction on NA12878.” (See page 8 paragraph 2)

Make Input to Network Clear

There are multiple points in the paper which make it seem as if the alignment information is not used as input to the network.

These sound like no alignment information is passed into the network.

Line 94 - 96 : “DeepMod takes fast5 files generated from Nanopore sequencer as inputs, and extracts signal information of each event to determine whether modifications occur. DeepMod

also summarizes whole-genome modification information after aligning long reads in fast5 files to a reference genome.”

Line 337-339 “In DeepMod, the input is FAST5 files generated by Nanopore sequencers with raw signals and base calls, and the output include modification prediction for each base in a long read and modification summary for each genomic position in a reference genome in BED format”

Line 345-346 “The input to DeepMod are raw signals and events in a FAST5 file generated by Nanopore sequencers with base calls”

Line 403-406 “In a real application, given a set of FAST5 files, DeepMod can generate prediction modification for each event in a long read, and a sequence of bases from events could be aligned with a reference genome. Then, a genomic position in a reference genome would be aligned with a set of events each from a long read and with a prediction modification label.”

This make it seem like the mapping information is passed into the network

Line 354-355 “Then, the signal mean, standard deviation, and the number of signals 355 associated with an event were extracted, plus a 4-vector feature for mapped reference bases.”

Input is not clear until this sentence.

Line 354-358 “Then, the signal mean, standard deviation, and the number of signals associated with an event were extracted, plus a 4-vector feature for mapped reference bases. In the 4-vector feature, 1 indicates the mapped reference base is a specific nucleotide type, while 0 mean otherwise. Thus, 7 features were used to describe an event (7-feature description for short and denoted by $x_i = [f_m, f_d, f_l, f_a, f_c, f_g, f_t]$.”

Response: Thank you for the comments. The input of the deep learning model in DeepMod is the summary of signals associated with an event and the mapped reference base type of the event. We have make the revision in the manuscript to avoid this confusion. Meanwhile, to make it more clear, we have now stated that the input of DeepMod includes a reference genome and fast5 files with raw signals and event information, and the input of the deep learning model in DeepMod is the summary information of an event and the mapped reference base type of the event (as Lines 354-358).

Make more calls on NA12878:

Try making calls on less confident CpG sites and see if proportion of calls matches the proportion of modification in the bisulfite sequencing data.

Response: According to your suggestion, we also made the prediction on CpG sites on NA12878 whose 5mC methylation percentage has small difference (<0.25) between the two replicates of bisulfite sequence (the methylation percentage could be from 0% to 100%) to reduce heterogeneity of sequencing data. We found that the correlation between the prediction methylation percentage and the percentage in bisulfite sequencing is >0.85 for majority of chromosomes when coverage ≥ 5. (page 8 paragraph 2 in the revised manuscript)

Make note that evaluation of R9.4 data with model trained from R9 data.

It sounds like a model trained with the E.coli R9 chemistry was able to predict accurately the NA12878 sequencing data from R9.4. If this is the case then that is worth speaking about. If this is the case it may be interesting to take the R9.4 model trained using NA12878 and make

inference calls on the E.coli data.

Supp Line 11-12 “This Nanopore sequencing dataset contains both positive control data and negative control data of E. coli. Both were sequenced by Simpson et. al. using Nanopore R9 sequencing techniques”

Line 182-184 “DeepMod 183 had the AP value equal to 0.998 and the AUC value to 0.995, indicating accurate prediction on human 184 genome, even when using a DeepMod model trained on E. coli data“

Response: According to your suggestion, we trained a model using completely methylated and completely un-methylated cytosines of chromosome 1 to 10 of NA12878 (where both NA12878 and E. coli data were basecalled with the same version of Albacore.). We then used DeepMod to make the prediction on Con1, Con2, GCGC_Hhal, CG_MpeI and CG_SssI (Stiober et. al. bioRxiv 2017). For each of three positive data (GCGC_Hhal or CG_MpeI or CG_SssI), we mixed motif-based 5mC with those motifs in both Con1 and Con2 for performance evaluation of DeepMod, and the number of the cytosines of interest in motifs from positive control and from Con1 and Con2 were shown in Table 2. With a coverage ≥ 1 , DeepMod achieved AP=0.95 on GCGC_Hhal, AP=0.807 on CG_MpeI, and AP=0.973 on CG_SssI. The AUC values achieved by DeepMod are 0.942, 0.809 and 0.965 on GCGC_Hhal, CG_MpeI and CG_SssI respectively. On GCGC_Hhal and CG_SssI. The AP values were 2-4 percent points decrease than the prediction by a DeepMod model trained on the E. coli data of CG_MSssI and UMR (Simpson et. al Nat Methods 2017), while the AP value (0.807) was 0.11 lower on CG_MpeI. In summary, the cross-organism testing suggested that DeepMod trained on one organism can make accurate on the other organisms, when the same basecalling algorithm was used. We note that the data from NA12878 was generated from native DNA while the data from Stiober et. al. bioRxiv 2017 were treated by PCR and enzymes: native DNAs may contain incompletely methylated cytosines due to cellular heterogeneity, which may result in slightly lower performance when training a model. (page 9 paragraph 2 in the revised manuscript)

Furthermore, in our analysis, we found that the different flow cell versions are not critical but the different Albacore versions can result in difference (even if the base calling results are similar between software versions, the extracted signal intensity can be quite different, which we need to use as input data)

Confused between, per-read, per-reference-site and per-call performance.

You seem to change the performance measurement between per-reference-site, per-read and per-call accuracy. It seems like the E.coli accuracy is measuring per-read methylation accuracy whereas the NA12878 accuracy is measuring aggregated per-reference-site accuracy.

In the supplementary tables you report per-read performance but I would want to see the performance on every called base. The classifier is only making one per-site prediction so aggregating calls into a per-read performance metric blurs the true accuracy of the classifier. Per-read accuracy depends on the length of the read and the number of called sites within that

read.

Response: Thank you for this comment. In the revised version, there are two ways to evaluate our tool: one is based on each genomic position of interest (per-reference-site), and the other is based on per call. The per-reference-site performance was used in the main manuscript to be consistent, and the per-call performance was only used in the supplementary tables 1 and 2 (we have revised Supplementary Table 1 for per-call performance rather than per-read performance).

-- Per read

Supp Line 119-121 - "To calculate per-read performance, each read would be evaluated individually with precision, recall, accuracy and F1-score on CG_MSsI and with accuracy on UMR"

-- Per-reference-site

Line 162-164 "We made 5mC prediction for each Nanopore long read, and each cytosine in a long read was associated with a prediction label indicating whether it was predicted to be modified or not. Then, after all long reads were aligned against a reference genome, predicted methylation summary for the whole genome was generated for each cytosine in the genome with a percentage to indicate how many modification predictions were made for that position, compared to all reads covering that position."

-- per-cal

Line 177-179: "We compared DeepMod prediction against the completely methylated and unmethylated bases for each chromosome"

E.coli was done per-genomic position?

Line 463-465: "Then, methylation prediction was summarized for each genomic position associated with a coverage, a methylation coverage and a methylation percentage."

NA12878 evaluation was done per call?

Lin 480-483 "Then, the DeepMod prediction was compared with this definition, and AP and AUC were calculated to evaluate binary classification performance of DeepMod."

Response: Thank you for this comment. In the revised version, we only have two kinds of performance: per-reference-site (genomic positions of interest) performance in the main text of the manuscript, and per-call performance for the Supplementary Tables 1 and 2. We have corrected all sentences to make the kinds of performance clearer.

Be specific about training, testing and validation numbers:

It is difficult to find out how many reads and individual sites/motifs were used for training for each model. Specifically how many sites were used for training, validation and test?

Response: For E. coli predictions, the number of testing sites were provided in Table 2, Figure 2(a)-(c) for 5mC and Figure 3(a) for 6mA, which is also available in the confusion matrix of Supplementary Table 3. The number of testing reads in human data was given in Supplementary Table 4 and 5. The training sites on E. coli for 5mC are similar to testing but with different Nanopore datasets, and the training sites on E coli for 6mA is ~4 times larger than the testing sites. The number of reads used for training are given in the first several paragraphs in the Supplementary file.

How long did it take to train?

How many iterations?

How many times did the model see each individual site?

Response: For *E. coli* data, 4 iteration of negative-control data was conducted, the model would see each read (not individual site) in negative control 4 times, and roughly 5 times for reads in positive control. The training took ~56-59 hours but IO consumed ~80% training times because large size of data.

What compute resources were used?

How long does it take to do inference on X number of reads on X number of GPU's/CPU's?

Response: During training process, GPU is used, while CPU is used for testing in a parallel way. On CPU prediction with 10 threads in a shared environment, roughly the prediction for a read took less than 1 second on average.

We would like to note that currently, DeepMod is available to run multiple jobs in parallel in a cluster environment and each with multiple threads, which could speed up the prediction process if enough resource is available. We also want to note that raw signal data of Nanopore sequencing is usually over several TB in size (in the case of human data, tens of TB), which can be time consuming to process due to I/O limitations.

What were your filtering steps for reads and for sites?

How many reads and sites were lost from each filter step?

Response: In the prediction stage, there is no filtering step for sites, and reads with mapped quality <10 would not be used (only tens of reads among thousands of passed reads had poor mapped quality in our experiments.). In the future, we can introduce quality control but quality control can also be done separately after DeepMod's prediction. We have discussed this point (page 17 and paragraph 1) in the revised manuscript.

Line 468-471: "All long reads of NAI2878 in FAST5 files together with a reference genome GRCh38 were used as input of DeepMod, and DeepMod made modification prediction for each bases in long reads with a label to indicate whether it was methylated."

Response: There is no filter step except tens of poorly mapped reads in this prediction, and after make the prediction for each bases of interest in long reads, the methylation information for a particular reference position of interest was summarized as stated in Lines after Line 468-471.

Limitations of Software:

Should mention how the software is not just limited by the type of modification but also by the

motif that the modification is found. Also, species-specificity is a concern. Whether there're some species-specific methylation contexts?

Line 250-252: "There are several current limitations in DeepMod that we wish to discuss here. First, DeepMod is trained and evaluated with 5mC and 6mA, two commonly available DNA modifications, and thus it might not work well with other types of modifications."

Response: We have added this motif limitation to the discussion according to your comment. (page 11 and paragraph 3) There might be species-specificity methylation contexts, but we have no enough data to evaluate this now. It would be a good topic when more methylation data from various species is available.

Also, once you note that nanopolish has a limitation in CpG sites, you should explain your limitation within the main paper. If the selection criteria for sites depends on the CIGAR string (which can change based on which aligner is used), it should be noted that re-alignments can bring in or exclude sites previously examined.

Line 197-199: "We note that nanopolish was designed for those CpG sites which were completely methylated/un-methylated CpG sites within 10bps and not semi-methylated, while DeepMod does not have this limitation"

Response: We have discussed all limitation in the discussion, and also added this limitation in the discussion according to your suggestion: "Fourth, DeepMod relied on alignment tool to find correct reference positions of bases in long reads. Now, DeepMod supports two widely-used aligners: BWA-MEM and minimap2. If different aligners or improper parameters are used with poor alignment performance, DeepMod's prediction would also be affected. Re-alignments with different tools or parameters might bring in new examined sites or exclude previously examined sites." (page 12 paragraph 1)

Supp Line 76-78 "Then, given a motif, for example, CpG sites, C in a CpG site was labelled to be methylated if (i) there were not more than 2 gaps in a 7-base window centered at that C, or (ii) there were not more than 3 gaps in a 13-base window centered at that C."

Response: The number of maximum gaps is used to eliminate poor alignment when infer methylation state of a base at a position in a long read. 13-base window is used to avoid the incorrect alignment due to methylations (methylation might results in wrong basecalling in a k-mer such as 5-mer) rather than due to alignment errors, while 7-base window is used to avoid other effect on the alignment. 7 is the minimum odd number larger than 5 and 5-mer is usually used for event in fast5 files, while 13 is almost twice of 7. These numbers is selected empirically as the default numbers, but can be changed by the users.

Error! Reference source not found:

There are several places where the reference to a table or citation is not found.

Response: Thank you. We have corrected them.

I only see 13 values of the 14 value vector for second deep neural network:

- 1: the predicted methylation percentage of a position*
- 2: corresponding position of CpG site at the reverse strand*
- 3: the number of nearby CpG sites within 25bp*
- 4-13: the 10 discretized bins*

Line 393-396: “ The input of this layer is a 14-value vector: the predicted methylation percentage of a position, that of corresponding position of CpG site at the reverse strand, the number of nearby CpG sites within 25bp, and the 10 discretized bins ([0, 0.05], [0.05, 0.15], [0.15, 0.25],.... [0.95, 1.0]) with value of how many percentage of nearby CpG sites has corresponding predicted methylation percentage. “

Response: The number of discretized bins would be 11 rather than 10. We have corrected this typos. Thank you.

REVIEWERS' COMMENTS:

Reviewer #1 (Remarks to the Author):

I think the manuscript has majorly improved and can now be published by Nature Communications.

Reviewer #2 (Remarks to the Author):

We (myself and lab members) read through and are satisfied with the revision. Good job!

Point-by-point response

We thank the reviewers and editors for your great evaluation of our manuscript titled “Detection of DNA base modifications by deep recurrent neural network on Oxford Nanopore sequencing data”. The point-by-point responses to the reviewers’ comments are provided below:

REVIEWERS' COMMENTS:

Reviewer #1 (Remarks to the Author):

I think the manuscript has majorly improved and can now be published by Nature Communications.

Response: Thank you very much for your positive comment and recommendation for publication.

Reviewer #2 (Remarks to the Author):

We (myself and lab members) read through and are satisfied with the revision. Good job!

Response: Thank you very much for all your great comments to improve our work and manuscript.